# A sharp Pif1-dependent threshold separates DNA double-strand breaks from critically short telomeres

Jonathan Strecker[1,2†], Sonia Stinus[3†], Mariana Pliego Caballero[3], Rachel K Szilard[1], Michael Chang[3]*, Daniel Durocher[1,2]*

[1]Lunenfeld-Tanenbaum Research Institute, Mount Sinai Hospital, Toronto, Canada; [2]Department of Molecular Genetics, University of Toronto, Toronto, Canada; [3]European Research Institute for the Biology of Ageing, University of Groningen, University Medical Center Groningen, Groningen, Netherlands

**Abstract** DNA double-strand breaks (DSBs) and short telomeres are structurally similar, yet they have diametrically opposed fates. Cells must repair DSBs while blocking the action of telomerase on these ends. Short telomeres must avoid recognition by the DNA damage response while promoting telomerase recruitment. In *Saccharomyces cerevisiae,* the Pif1 helicase, a telomerase inhibitor, lies at the interface of these end-fate decisions. Using Pif1 as a sensor, we uncover a transition point in which 34 bp of telomeric $(TG_{1-3})_n$ repeat sequence renders a DNA end insensitive to Pif1 action, thereby enabling extension by telomerase. A similar transition point exists at natural chromosome ends, where telomeres shorter than ~40 bp are inefficiently extended by telomerase. This phenomenon is not due to known Pif1 modifications and we instead propose that Cdc13 renders $TG_{34+}$ ends insensitive to Pif1 action. We contend that the observed threshold of Pif1 activity defines a dividing line between DSBs and telomeres.

DOI: https://doi.org/10.7554/eLife.23783.001

*For correspondence:
m.chang@umcg.nl (MC);
durocher@lunenfeld.ca (DD)

†These authors contributed equally to this work

Competing interests: The authors declare that no competing interests exist.

## Introduction

A fundamental question in chromosome biology is how cells differentiate between DNA double-strand breaks (DSBs) and telomeres, the natural ends of chromosomes. A failure to distinguish between these structures has severe consequences for genome integrity. For example, the engagement of the non-homologous end-joining pathway at telomeres can lead to breakage-fusion-bridge cycles that wreak havoc on the genome. Similarly, the activity of telomerase at DSBs can generate new telomeres at the cost of the genetic information distal to the break. Telomere addition has been observed in a variety of species (*Kramer and Haber, 1993*; *Biessmann et al., 1990*; *Fouladi et al., 2000*) and has been linked to human disorders involving terminal deletions of chromosome 16 (*Wilkie et al., 1990*) and 22 (*Wong et al., 1997*). While DSBs and telomeres reflect extreme positions on the spectrum, a continuum of DNA ends exists between them, including critically short telomeres and DSBs occurring in telomeric-like sequence. All these require a decision: should the end be repaired or should it be elongated by telomerase?

The budding yeast *Saccharomyces cerevisiae,* whose telomeres consist of 300 ± 75 bp of heterogeneous $(TG_{1-3})_n$ repeats, has been a key model to study mechanisms of genomic stability (*Zakian, 1996*). The telomere repeats organize a nucleoprotein structure minimally composed of the double-stranded (ds) DNA binding protein Rap1, its interacting factors Rif1 and Rif2, and the telomere-specific single-stranded (ss) DNA-binding Cdc13-Stn1-Ten1 (CST) complex, which caps the chromosome ends (*Dewar and Lydall, 2012*). These telomere-bound proteins prevent activation of DNA damage signaling pathways and the ability of the DSB repair machinery to use telomeric ends

as substrates. This so-called capping function is a universal property of eukaryotic telomeres; while different in composition, a set of human proteins collectively known as shelterin accomplishes a similar function in human cells (*Palm and de Lange, 2008*).

Telomerase-mediated extension does not occur at every telomere in every cell cycle, but the probability of telomere extension steadily increases as telomere length decreases (*Teixeira et al., 2004*). Telomerase also acts more processively at telomeres less than 125 bp in length, resulting in more extensive elongation of critically short telomeres (*Chang et al., 2007*). The preferential extension of short telomeres can be rationalized since short telomeres are most in danger of becoming dysfunctional. Thus, while telomerase must be tightly inhibited at DSBs, its activity must also be suppressed at telomeres that are sufficiently long. A number of proteins have been implicated in this process, including Rif1, Rif2, and the Tel1 (ATM) kinase (reviewed in *Wellinger and Zakian, 2012*). In addition, the telomerase inhibitor Pif1, which is a helicase that unwinds RNA-DNA hybrids in vitro and removes telomerase from telomeric DNA (*Boulé et al., 2005*), has recently been shown to act preferentially at long telomeres (*Phillips et al., 2015*).

Remarkably, Pif1 is also required to inhibit telomerase at DSBs. Pif1 has both mitochondrial and nuclear isoforms encoded from separate translational start sites; mutation of the second start site in the *pif1-m2* mutant abolishes the nuclear isoform, resulting in telomere elongation (*Schulz and Zakian, 1994*) and a 240-fold increase in telomere addition at DSBs (*Myung et al., 2001*). The Mec1 (ATR)-dependent phosphorylation of Cdc13 also guards against the inappropriate recruitment of the CST complex to DSB sites (*Zhang and Durocher, 2010*).

One striking feature of Pif1 is that it is able to distinguish between DSBs and telomeres, as a *pif1-4A* mutant affects telomere addition frequency at DSBs but without influencing native telomere length (*Makovets and Blackburn, 2009*). This observation makes Pif1 an attractive candidate for a protein that controls the distinction between DSBs and short telomeres. We noted in our previous work that Pif1 suppresses telomere addition at HO-induced DSBs containing 18 bp of $(TG_{1-3})_n$ telomeric repeats (referred to as $TG_{18}$) but has no impact on the telomerase-dependent elongation of DNA ends containing a $TG_{82}$ sequence (*Zhang and Durocher, 2010*). This observation suggests that the $TG_{82}$ substrate behaves as a critically short telomere and that cells elongate it in a manner that is uninhibited by Pif1. Thus, this system appears to recapitulate the end-fate decisions undertaken at DSBs versus critically short telomeres.

## Results

### Identification of a Pif1-insensitivity threshold at DNA ends

To characterize the dividing line between a DSB and a short telomere, we used a genetic system in which galactose-inducible HO endonuclease can be expressed to create a single DSB at the *ADH4* locus on Chr VII-L (*Diede and Gottschling, 1999*; *Gottschling et al., 1990*). By placing different lengths of telomeric $(TG_{1-3})_n$ sequence immediately adjacent to the HO cut site, one can study the fate of DNA ends using two readouts: a genetic assay for telomere addition based on the loss of the distal *LYS2* marker, and by Southern blotting to monitor the length of the DNA end (*Figure 1a,b*). The HO cut site in this system contributes one thymine nucleotide to the inserted telomeric seed, accounting for a one base pair discrepancy from prior reports. As previous work indicated that Pif1 is active at $TG_{18}$, but not $TG_{82}$ (*Zhang and Durocher, 2010*), we first constructed strains containing 34, 45, 56, and 67 bp of telomeric repeats in both wild-type and *pif1-m2* cells (see *Supplementary file 1A* for all TG repeat sequences). We observed similar rates of telomere addition at all DNA ends in both backgrounds, indicating that 34 bp of telomeric repeats are sufficient to render a DNA end insensitive to Pif1 (*Figure 1c*; source data are available in *Figure 1—source data 1*). To account for variations in HO cutting efficiency and the propensity to recruit telomerase at each DNA end, we also normalized telomere addition frequency to *pif1-m2* cells to provide a clear readout of Pif1 activity (*Figure 1—figure supplement 1a*; source data are available in *Figure 1—figure supplement 1—source data 1*). Analysis of DNA ends by Southern blot also revealed robust telomere addition at the $TG_{34}$ substrate in *PIF1* cells, mirroring the results of the genetic assay (*Figure 1d*).

The standard genetic telomere addition assay includes a nocodazole arrest before DSB induction, as telomerase is active in S/G2 phase (*Diede and Gottschling, 1999*). However, asynchronously

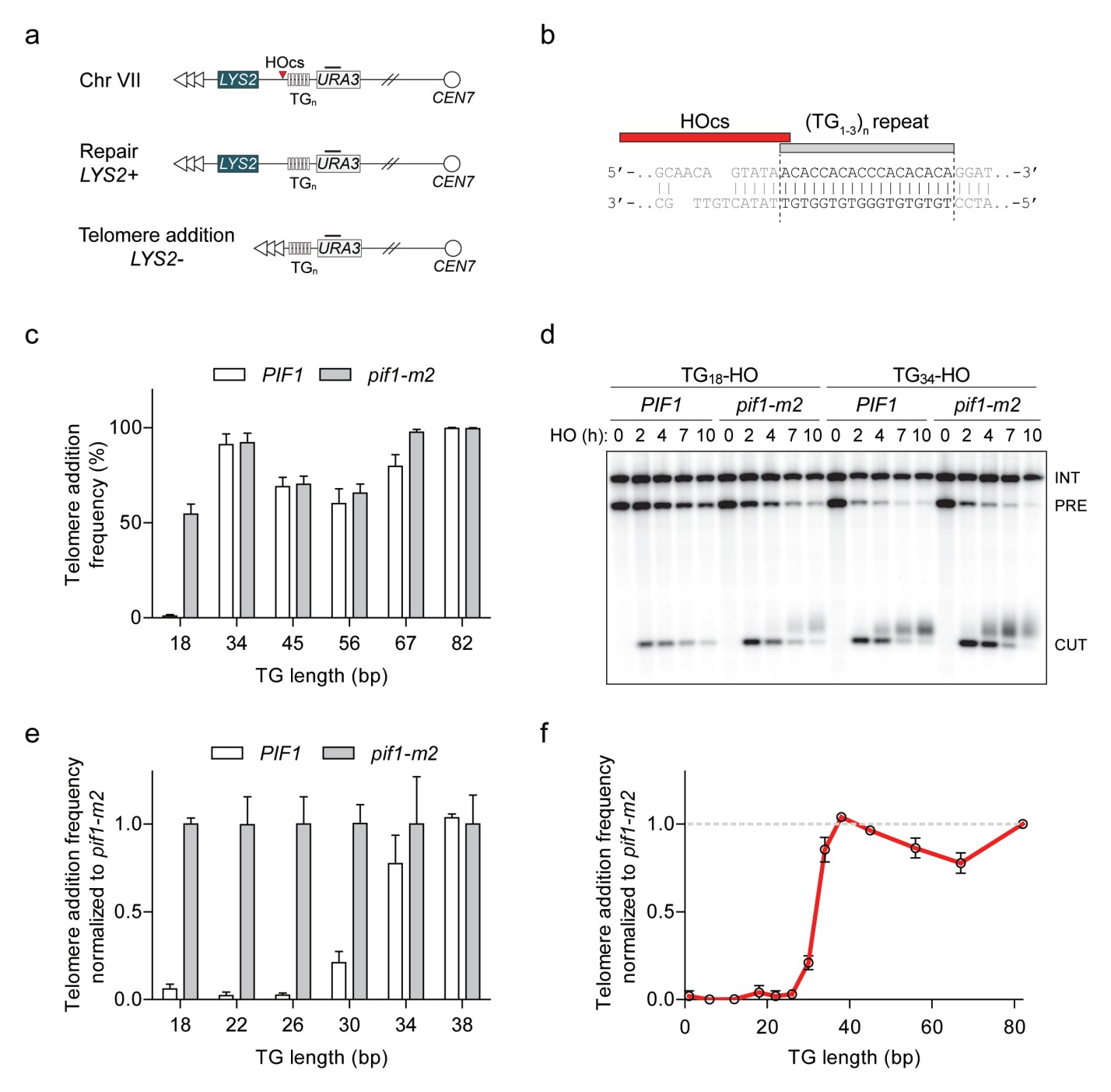

**Figure 1.** Characterization of Pif1 activity at DNA ends reveals a DSB-telomere transition. (**a**) Schematic of a system to study the fate of DNA ends. Telomeric repeats are placed adjacent to an HO cut site (HOcs) at the *ADH4* locus on Chr VII. Telomere addition can be measured using a genetic assay based on the loss of the distal *LYS2* gene as measured by resistance to α-aminoadipic acid. Southern blotting with a probe complementary to *URA3* (black bar) allows for visualization of DNA end stability. (**b**) Sequence of the $TG_{18}$ substrate and the overhang produced by the HO endonuclease. The C-rich strand runs 5′ to 3′ toward the centromere and is resected following DSB induction to expose a 3′ G-rich overhang. (**c**) Telomere addition frequency at DNA ends containing 18–82 bp of TG sequence. Data represent the mean ± s.d. from a minimum of n = 3 independent experiments. See ***Supplementary file 1a*** for the sequences of all DNA ends and source data are found in ***Figure 1—source data 1***. (**d**) Southern blot of DNA ends containing $TG_{18}$ and $TG_{34}$ ends in wild-type and *pif1-m2* cells following HO induction. A *URA3* probe was used to label the *ura3-52* internal control (INT) and the *URA3* gene adjacent to the $TG_n$-HO insert (PRE) which is cleaved by HO endonuclease (CUT). Newly added telomeres are visualized as a heterogeneous smear above the CUT band. (**e**) Telomere addition frequency normalized to *pif1-m2* cells at DNA ends containing 18–38 bp of TG

*Figure 1 continued on next page*

*Figure 1 continued*

sequence. Data represent the mean ± s.d. from n = 3 independent experiments. Source data are found in **Figure 1—source data 2**. (f) Summary of telomere addition frequency normalized to *pif1-m2* across the spectrum of TG repeat substrates. Source data are found in **Figure 1—source data 3**.
DOI: https://doi.org/10.7554/eLife.23783.002

The following source data and figure supplements are available for figure 1:

**Source data 1.** Raw data for telomere addition assays shown in *Figure 1c*.
DOI: https://doi.org/10.7554/eLife.23783.004

**Source data 2.** Raw data for telomere addition assays shown in *Figure 1e*.
DOI: https://doi.org/10.7554/eLife.23783.005

**Source data 3.** Raw data telomere addition assays shown in *Figure 1f*.
DOI: https://doi.org/10.7554/eLife.23783.006

**Figure supplement 1.** Characterizing a threshold of Pif1 activity at DNA ends.
DOI: https://doi.org/10.7554/eLife.23783.003

**Figure supplement 1—source data 1.** Raw data telomere addition assays shown in *Figure 1—figure supplement 1a*.
DOI: https://doi.org/10.7554/eLife.23783.007

**Figure supplement 1—source data 2.** Raw data for telomere addition assays shown in *Figure 1—figure supplement 1b*.
DOI: https://doi.org/10.7554/eLife.23783.008

**Figure supplement 1—source data 3.** Raw data for telomere addition assays shown in *Figure 1—figure supplement 1c*.
DOI: https://doi.org/10.7554/eLife.23783.009

dividing cells also exhibited a similar phenotype at the $TG_{18}$ and $TG_{34}$ ends (***Figure 1—figure supplement 1b***; source data are available in ***Figure 1—figure supplement 1—source data 2***). To exclusively study telomere addition by telomerase and not through homologous recombination, telomere addition strains also harbored a *rad52Δ* mutation. The addition of *RAD52* in this assay reduced telomere addition at the $TG_{18}$ end in *pif1-m2* cells but had no impact on the behavior of Pif1 at the $TG_{34}$ substrate (***Figure 1—figure supplement 1b***).

To further refine the Pif1-insensitivity threshold, we added 4 bp increments of TG repeat sequence to the centromeric side of the $TG_{18}$ substrate yielding strains with 22, 26, 30, 34, and 38 bp of telomeric repeats. Importantly, with the exception of length, these strains contained the same DNA sequence and shared a common distal end. Analysis of telomere addition revealed that Pif1 is active at DNA ends up to $TG_{26}$, while the frequency of telomere addition increased at the $TG_{30}$ end and beyond (***Figure 1e***; source data are available in ***Figure 1—source data 2***). As telomeric repeats are heterogeneous in nature, we next determined if this phenotype is dependent on the particular DNA sequence. We selected three different sequences from *S. cerevisiae* telomeric DNA and constructed strains with DNA ends containing either 26 or 36 bp of each sequence. Consistent with our initial observations, telomere addition was inhibited by Pif1 at all $TG_{26}$ ends, while the corresponding $TG_{36}$ ends resulted in telomere addition in the presence of Pif1 (***Figure 1—figure supplement 1c,d***; source data for panel c are available in ***Figure 1—figure supplement 1—source data 3***).

Visualization of the combined genetic assay results across different lengths of TG-repeat substrates reveals a striking transition with regard to Pif1 function (***Figure 1f***; source data are available in ***Figure 1—source data 3***). By using Pif1-insensitivity as an operational definition of a short telomere, we propose that the 26 to 34 bp window of telomeric sequence is the dividing line between what the cell interprets to be a DSB and what is considered to be a critically short telomere. These data suggest that DNA ends containing 34 bp or more of telomeric DNA are allowed to elongate in a manner unimpeded by Pif1 and we herein refer to this phenomenon as the DSB-telomere transition.

## A DSB-telomere transition also exists at chromosome ends

To validate the threshold that defines the DSB-telomere transition, we set up a system based on the STEX (Single Telomere EXtension) assay to monitor telomerase-mediated extension events at chromosome ends at nucleotide resolution after a single cell cycle (***Teixeira et al., 2004***). In the STEX assay, a clonal population of telomerase-negative cells is mated to a strain expressing telomerase. Telomeres that had shortened in the telomerase-negative cells can then be re-extended in the zygote. DNA is isolated from the zygotes and telomere elongation can be detected by amplifying, cloning and sequencing telomeres originating from the telomerase-negative strain. Since yeast

telomerase adds imperfect 5′-(TG)$_{0-6}$TGGGTGTG(G)$_{0-1}$-3′ repeats (**Förstemann and Lingner, 2001**), telomere elongation can be detected after sequence alignment of the telomeres because newly added sequences do not align with the non-elongated telomeres. We call these newly added sequences 'sequence divergence events' because they diverge from the original sequences. We introduced two major modifications to the STEX assay: (1) we use a strain where the expression of *EST1*, encoding a subunit of telomerase (**Lundblad and Szostak, 1989**), is under the control of a galactose-inducible promoter, allowing us to avoid the challengingly high mating efficiency needed in the classical STEX assay; (2) we make use of a *tlc1* template mutant (*tlc1-tm*) that introduces 5′-[(TG)$_{0-4}$TGG]$_n$ATTTGG-3′ telomeric repeats (**Chang et al., 2007**), enabling us to distinguish sequence divergence events that are telomerase-dependent (i.e. the divergent sequence is mutant) from those that are telomerase-independent (i.e. the divergent sequence is wild type). This modification was found to be important since a fraction of sequence divergence events can occur due to homologous recombination, as well as from errors introduced during amplification, cloning and sequencing of the telomeres (**Claussin and Chang, 2016**). Importantly, our iSTEX (for inducible STEX) data are similar to previously published STEX data (**Teixeira et al., 2004**; **Arnerić and Lingner, 2007**; **Ji et al., 2008**) in terms of the frequency and extent of telomere elongation events, and use of the *tlc1-tm* mutant does not significantly affect the repeat addition processivity of telomerase (**Chang et al., 2007**).

In this revised assay, we transform a PCR fragment containing the *tlc1-tm* allele into a strain with *EST1* under the control of a galactose-inducible promoter (**Figure 2a**). From the moment we transform strains with the *tlc1-tm* PCR fragment, we keep the cells in media containing glucose, which shuts off *EST1* expression and causes the telomeres to shorten. We then arrest successfully transformed cells in late G1 phase and release them in the presence of galactose to reactivate telomerase, allowing the addition of mutant sequences to the chromosome ends. We monitor the arrest/release efficiency by flow cytometry (**Figure 2b**), extract genomic DNA from released cells that have completed DNA replication, amplify telomeres by PCR, and then clone and sequence telomeres.

We monitor telomere sequence addition at an engineered V-R telomere, which contains an *ADE2* marker placed adjacent to the telomere repeats (**Singer and Gottschling, 1994**), and at the natural VI-R telomere. In agreement with previous reports (**Teixeira et al., 2004**), there is a strong preference to elongate short telomeres [(**Figure 2c**; source data are available in **Figure 2—source data 1**) and (**Figure 2—figure supplement 1a**; source data are available in **Figure 2—figure supplement 1—source data 1**)] and the frequency of telomerase-independent sequence divergence events is similar to previous reports where telomerase is knocked out (**Teixeira et al., 2004**; **Chang et al., 2011**; **Claussin and Chang, 2016**). These data indicate that the presence of telomerase does not influence these events. Strikingly, at both the V-R and VI-R telomeres, the frequency of telomere extension drops dramatically at extremely short telomeres (**Figure 2c**, **Figure 2—figure supplement 1a**). At the V-R telomere, only two out of 32 telomeres (6.3%) shorter than 44 bp were extended by telomerase, while 65 out of 136 telomeres (47.8%) between 44 bp and 86 bp long were extended. Similarly, at the VI-R telomere, two of the 13 telomeres below 38 bp (15.4%) were extended, while 51 out of 115 telomeres (44.3%) between 38 bp and 74 bp long were extended. Thus, while telomerase preferentially elongates short telomeres, those below ~40 bp are inefficiently extended. These data suggest that the DSB-telomere transition identified at HO-induced breaks also exists at native chromosome ends.

To determine whether Pif1 is also important for the DSB-telomere transition at chromosome ends, the iSTEX assays were repeated in a *pif1-m2* background [(**Figure 2d**; source data are available in **Figure 2—source data 2**) and (**Figure 2—figure supplement 1b**; source data are available in **Figure 2—figure supplement 1—source data 2**)]. At both the V-R and VI-R telomeres, the percentage of elongated telomeres below the DSB-telomere transition length determined in *PIF1* cells increased in the *pif1-m2* background (although statistical significance was not reached for the VI-R telomere due to difficulties in obtaining enough short telomeres for analysis). Thus, at both DSBs and chromosome ends, Pif1 is needed to set the DSB-telomere transition.

## Pif1 is not inhibited by DNA damage kinases

One attractive mechanism for the observed DSB-telomere transition is that Pif1 might be inactivated at DNA ends containing telomeric repeats 34 bp in length or longer. Prime candidates for this

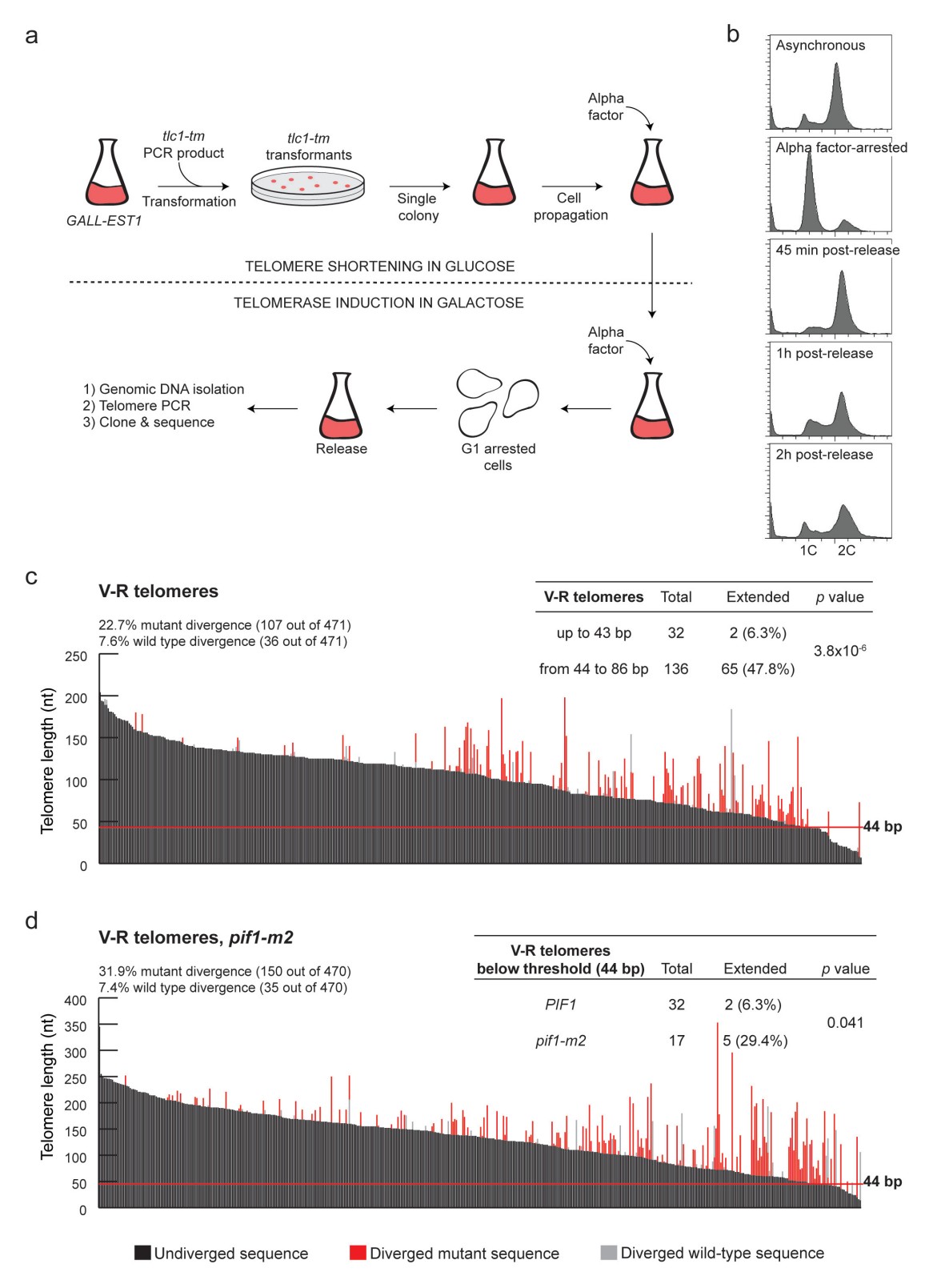

**Figure 2.** Characterization of the DSB-telomere transition at chromosome ends. (a) Methodology of the iSTEX assay; see text for details. (b) Arrest and release efficiency monitored by flow cytometry. (c) Telomere V-R was amplified, cloned and sequenced after 2 hr of *tlc1-tm* telomerase induction. Each bar represents an individual telomere. The black portion of each bar represents the undivergent sequence, the red portion shows the mutant divergent sequence and the grey portion indicates the wild-type divergent sequence. Telomeres are sorted based on the length of the undiverged sequence

*Figure 2 continued on next page*

*Figure 2 continued*

(black portion). The horizontal red line indicates the threshold below which telomerase-mediated telomere extension becomes very inefficient. Statistical analysis was done using a Fisher's exact test and for this analysis, telomeres containing only wild-type divergence were excluded. Source data are found in *Figure 2—source data 1*. (d) As in panel (c) except in a *pif1-m2* background. Statistical analysis comparing extension of telomeres below 44 bp in length in wild type and *pif1-m2* was done using a Fisher's exact test. Source data are found in *Figure 2—source data 2*.

DOI: https://doi.org/10.7554/eLife.23783.010

The following source data and figure supplements are available for figure 2:

**Source data 1.** Raw data for iSTEX shown in *Figure 2c*.
DOI: https://doi.org/10.7554/eLife.23783.012
**Source data 2.** Raw data for iSTEX shown in *Figure 2d*.
DOI: https://doi.org/10.7554/eLife.23783.013
**Figure supplement 1.** Characterization of the DSB-telomere transition at telomere VI-R.
DOI: https://doi.org/10.7554/eLife.23783.011
**Figure supplement 1—source data 1.** Raw data for iSTEX shown in *Figure 2—figure supplement 1a*.
DOI: https://doi.org/10.7554/eLife.23783.014
**Figure supplement 1—source data 2.** Raw data for iSTEX shown in *Figure 2—figure supplement 1b*.
DOI: https://doi.org/10.7554/eLife.23783.015

regulation include the central DNA damage kinases including Mec1 (ATR), Tel1 (ATM), and Rad53 (CHK2). Previous work has identified that Tel1 promotes telomerase-mediated extension of a $TG_{82}$ end (*Frank et al., 2006*), and targets short telomeres for elongation (*Sabourin et al., 2007*; *Hector et al., 2007*; *Arnerić and Lingner, 2007*). As these results raised the possibility that Tel1 antagonizes Pif1, we deleted *TEL1* in wild-type and *pif1-m2* backgrounds and followed the fate of the $TG_{82}$ DNA end by Southern blotting. Although telomere addition was reduced in *tel1Δ* cells, we observed a similar reduction in *tel1Δ pif1-m2* cells, indicating that *TEL1* and *PIF1* function in separate pathways (*Figure 3a,b*; source data for panel b are available in *Figure 3—source data 1*). Consistent with this observation, the loss of *TEL1* did not affect the DSB-telomere transition at the $TG_{18}$ and $TG_{34}$ DNA ends (*Figure 3c*; source data are available in *Figure 3—source data 2*). Loss of *MEC1* and *RAD53* also failed to inhibit telomerase in a Pif1-specific manner at the $TG_{82}$ end (*Figure 3—figure supplement 1a–d*; source data are available in *Figure 3—figure supplement 1—source data 1*, *2*). Pif1 contains five consensus S/T-Q Mec1 and Tel1 phosphorylation sites; however, their mutation in the *pif1-5AQ* allele (S148A/S180A/T206A/S707A/T811A) also did not decrease telomere addition at the $TG_{34}$ end (*Figure 3d*; source data are available in *Figure 3—source data 3*).

As Pif1 might be regulated through unanticipated post-translational modifications or protein-protein interactions, we performed a *PIF1* PCR mutagenesis screen to identify gain-of-function mutations that inhibit telomere addition at the $TG_{82}$ end but we were unable to recover any mutants. Together, these data suggest that Pif1 is not directly inactivated at the $TG_{34}$ and $TG_{82}$ DNA ends, so we next considered alternative explanations for the observed DSB-telomere transition.

## Artificial telomerase recruitment does not outcompete Pif1

A simple explanation for the DSB-telomere transition is that longer telomeric repeats might have an increased ability to recruit telomerase. This model predicts that artificially increasing telomerase recruitment to the $TG_{18}$ end might be sufficient to overcome Pif1 inhibition. Since the primary mechanism of telomerase recruitment involves an interaction between the DNA-binding protein Cdc13 and the Est1 telomerase subunit (*Nugent et al., 1996*; *Pennock et al., 2001*), we expressed Cdc13-Est1 and Cdc13-Est2 fusion proteins (*Evans and Lundblad, 1999*) to test this possibility. In agreement with previous work, expression of both fusions resulted in greatly elongated telomeres (*Evans and Lundblad, 1999*) (*Figure 4a*); however, they did not increase telomere addition at the $TG_{18}$ DNA end in the presence of Pif1 (*Figure 4b*; source data are available in *Figure 4—source data 1*). To test whether the Cdc13-Est1 fusion protein is able to bind and extend the $TG_{18}$ substrate, we repeated the genetic telomere healing assay in *est1Δ* cells expressing a Cdc13-Est1 fusion containing the *est1-60* mutation (K444E) that disrupts the interaction of Est1 with endogenous Cdc13 (*Pennock et al., 2001*). Telomerase extension in these *est1Δ* cells must therefore arise from the ectopic construct. We observed that Cdc13-Est1$^{K444E}$ can extend the $TG_{18}$ end only in the absence of *PIF1* (*Figure 4b*). Together, these data indicate that Pif1 is able to effectively suppress

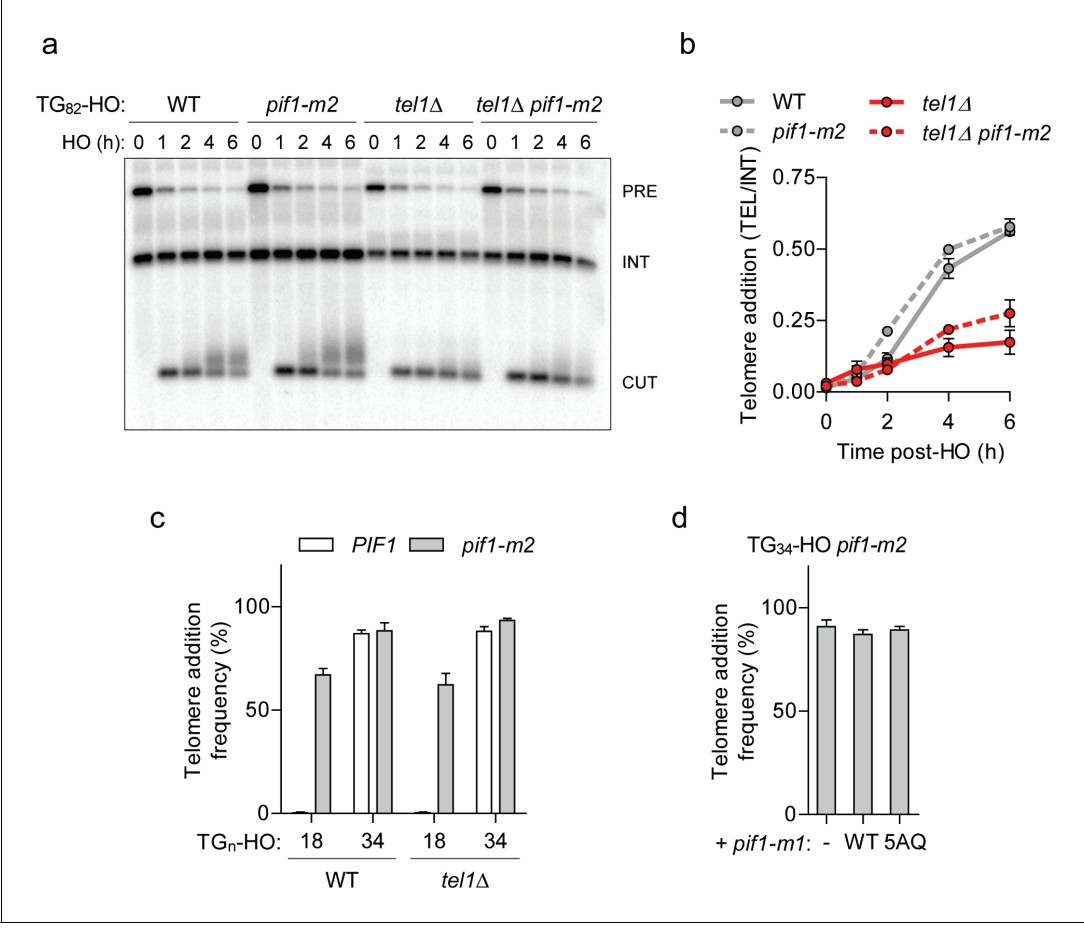

**Figure 3.** Pif1 is not inactivated by Tel1 at short telomeres. (**a, b**) Southern blot of the $TG_{82}$ DNA end following HO-induction in wild-type (WT) and *pif1-m2* cells without or with a *TEL1* deletion. An *ADE2* probe was used to label the *ade2Δ1* internal control (INT) and the *ADE2* gene adjacent to the $TG_n$-HO insert (PRE) which is cleaved by HO endonuclease (CUT). Quantification of the newly added telomere signal (**b**) calculated by subtracting the background signal present prior to HO-induction and by normalizing to the INT control. Data represent the mean ± s.d. from n = 2 independent experiments. Source data are found in *Figure 3—source data 1*. (**c**) Telomere addition frequency at the $TG_{18}$ and $TG_{34}$ DNA ends in *tel1Δ* mutants. Data represent the mean ± s.d. from n = 3 independent experiments. Source data are found in *Figure 3—source data 2*. (**d**) Telomere addition frequency at the $TG_{34}$ DNA end in *pif1-m2* cells (-) and cells expressing a wild-type (WT) or *pif1-5AQ* (S148A, S180A, T206A, S707A, T811A) nuclear-specific *pif1-m1* allele. Data represent the mean ± s.d. from n = 3 independent experiments. The functionality of the *pif1-m1* alleles was assessed by rescue of the telomere elongation associated with *pif1-m2* (*Figure 3—figure supplement 1e*). Source data are found in *Figure 3—source data 3*.
DOI: https://doi.org/10.7554/eLife.23783.016

The following source data and figure supplements are available for figure 3:

**Source data 1.** Raw data for telomere addition assays shown in *Figure 3b*.
DOI: https://doi.org/10.7554/eLife.23783.018

**Source data 2.** Raw data for telomere addition assays shown in *Figure 3c*.
DOI: https://doi.org/10.7554/eLife.23783.019

**Source data 3.** Raw data for telomere addition assays shown in *Figure 3d*.
DOI: https://doi.org/10.7554/eLife.23783.020

**Figure supplement 1.** Loss of *MEC1* or *RAD53* does not affect Pif1 at short telomeres.
DOI: https://doi.org/10.7554/eLife.23783.017

**Figure supplement 1—source data 1.** Raw data for telomere addition assays shown in *Figure 3—figure supplement 1b*.
DOI: https://doi.org/10.7554/eLife.23783.021

**Figure supplement 1—source data 2.** Raw data for telomere addition assays shown in *Figure 3—figure supplement 1d*.
DOI: https://doi.org/10.7554/eLife.23783.022

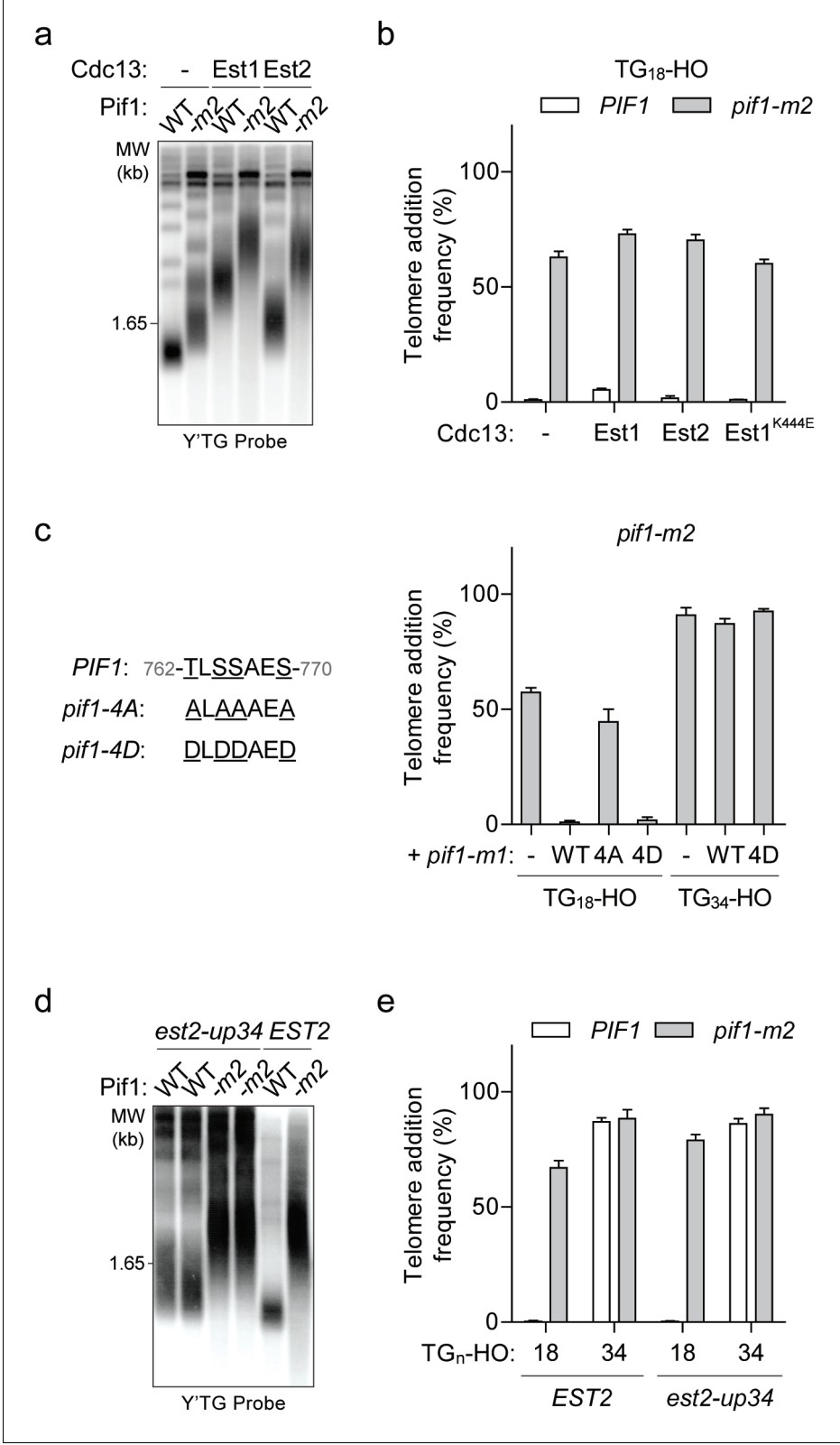

**Figure 4.** The DSB-telomere transition recapitulates the differential regulation of Pif1. (**a**) Southern blot for telomere length in $TG_{18}$-HO wild-type (WT) and *pif1-m2* cells harbouring an empty plasmid (-) or expressing plasmid-based Cdc13-Est1 or Cdc13-Est2 fusions. Cells were passaged for approximately 75 generations before genomic DNA extraction. A Y′ TG probe was used to label telomeric sequences. (**b**) Telomere addition frequency
*Figure 4 continued on next page*

*Figure 4 continued*

of the cells described in panel (a) and *est1Δ* strains expressing a *cdc13-est1-60* (K444E) fusion. Data represent the mean ± s.d. from n = 3 independent experiments. Source data are found in *Figure 4—source data 1*. (c) Telomere addition frequency at the $TG_{18}$ and $TG_{34}$ DNA ends in *pif1-m2* cells (-) and cells expressing a wild-type (WT), *pif1-4A* (T763A/S765A/S766A/S769A), or *pif1-4D* (T763A/S765A/S766A/S769A) nuclear-specific *pif1-m1* allele. Data represent the mean ± s.d. from n = 3 independent experiments. Source data are found in *Figure 4—source data 2*. (d) Southern blot for telomere length in *PIF1* (WT) and *pif1-m2* cells combined with or without the *est2-up34* mutation. Cells were passaged for approximately 75 generations before genomic DNA extraction and a Y' TG probe was used to label telomere sequences. (e) Telomere addition frequency at the $TG_{18}$ and $TG_{34}$ DNA ends in *PIF1* and *pif1-m2* cells with or without the *est2-up34* mutation. Data represent the mean ± s.d. from n = 3 independent experiments. The *EST2* data are the same as the WT data shown in *Figure 3c*. Source data are found in *Figure 4—source data 3*.

DOI: https://doi.org/10.7554/eLife.23783.023

The following source data is available for figure 4:

**Source data 1.** Raw data for telomere addition assays shown in *Figure 4b*.
DOI: https://doi.org/10.7554/eLife.23783.024
**Source data 2.** Raw data for telomere addition assays shown in *Figure 4c*.
DOI: https://doi.org/10.7554/eLife.23783.025
**Source data 3.** Raw data for telomere addition assays shown in *Figure 4e*.
DOI: https://doi.org/10.7554/eLife.23783.026

---

telomere addition even in the presence of enhanced telomerase recruitment, suggesting that increased telomerase recruitment to the $TG_{34}$ end is unlikely to underpin the observed DSB-telomere transition.

## The DSB-telomere transition recapitulates the differential regulation of Pif1

Pif1 may only be activated at DNA ends with short tracts of telomeric sequence. Consistent with this model, Pif1 is reported to be phosphorylated after DNA damage in a Mec1-Rad53-Dun1-dependent manner and further characterization of this activity led to the identification of the *pif1-4A* mutant (T763A/S765A/S766A/S769A) that is unable to inhibit telomere addition at DSBs (*Makovets and Blackburn, 2009*). Importantly, mimicking phosphorylation with the *pif1-4D* allele can restore Pif1 activity (*Makovets and Blackburn, 2009*). We first confirmed the function of these mutants at the $TG_{18}$ DNA end by integrating variants of the nuclear-specific *pif1-m1* allele at the *AUR1* locus in *pif1-m2* cells (*Figure 4c*; source data are available in *Figure 4—source data 2*). As expected, introducing the *pif1-m1* allele rescued the telomere lengthening phenotype of the *pif1-m2* allele (*Figure 3—figure supplement 1e*). If Pif1 phosphorylation only occurs at DNA ends with short lengths of telomeric repeats, such as $TG_{18}$, then mimicking phosphorylation may be sufficient to inhibit telomere addition at DNA ends with longer telomeric repeats. Contrary to this prediction, the *pif1-4D* mutant did not restore Pif1 activity at the $TG_{34}$ DNA end (*Figure 4c*), indicating that phosphorylation of these sites is not sufficient to regulate the DSB-telomere transition.

Several lines of evidence indicate that Pif1 functions differently at DSBs and telomeres. First, the *pif1-4A* mutation affects the frequency of telomere addition at DSBs, but does not affect native telomere length (*Makovets and Blackburn, 2009*). The inability of the *pif1-4D* allele to inhibit telomerase at $TG_{34}$ therefore provides indirect evidence that the cell interprets this DNA end as a short telomere. A second mutation that affects Pif1 activity has also been identified: the *est2-up34* mutation, which affects the finger domain of the telomerase reverse transcriptase subunit (*Eugster et al., 2006*). Interestingly, the *est2-up34* mutant results in over-elongated telomeres in wild-type but not *pif1-m2* cells, indicating that the *est2-up34* allele can at least partially bypass Pif1 inhibition (*Eugster et al., 2006*). To test if this holds true at DSBs, we generated the *est2-up34* mutation in strains with a $TG_{18}$ DNA end. Although we observed increased telomere length in *PIF1 est2-up34* cells (*Figure 4d*), telomere addition was not increased (*Figure 4e*; source data are available in *Figure 4—source data 3*), indicating that the *est2-up34* mutation can bypass Pif1 function at telomeres but not at DSBs. Together these data support the idea that Pif1 possesses distinct functions at DSBs

and telomeres and that these differences are recapitulated in the $TG_{18}$ and $TG_{34}$ DNA ends on either side of the DSB-telomere transition.

## Investigating the molecular trigger of the DSB-telomere transition

Since our attempts thus far failed to identify a modification of Pif1 that would explain the DSB-telomere transition, we next focused on whether a property of the DNA end facilitates or blocks Pif1 activity. Attractive candidates included the MRX and Ku complexes, which are rapidly recruited to DNA ends and function in both DSB repair and telomere maintenance (*Dewar and Lydall, 2012*).

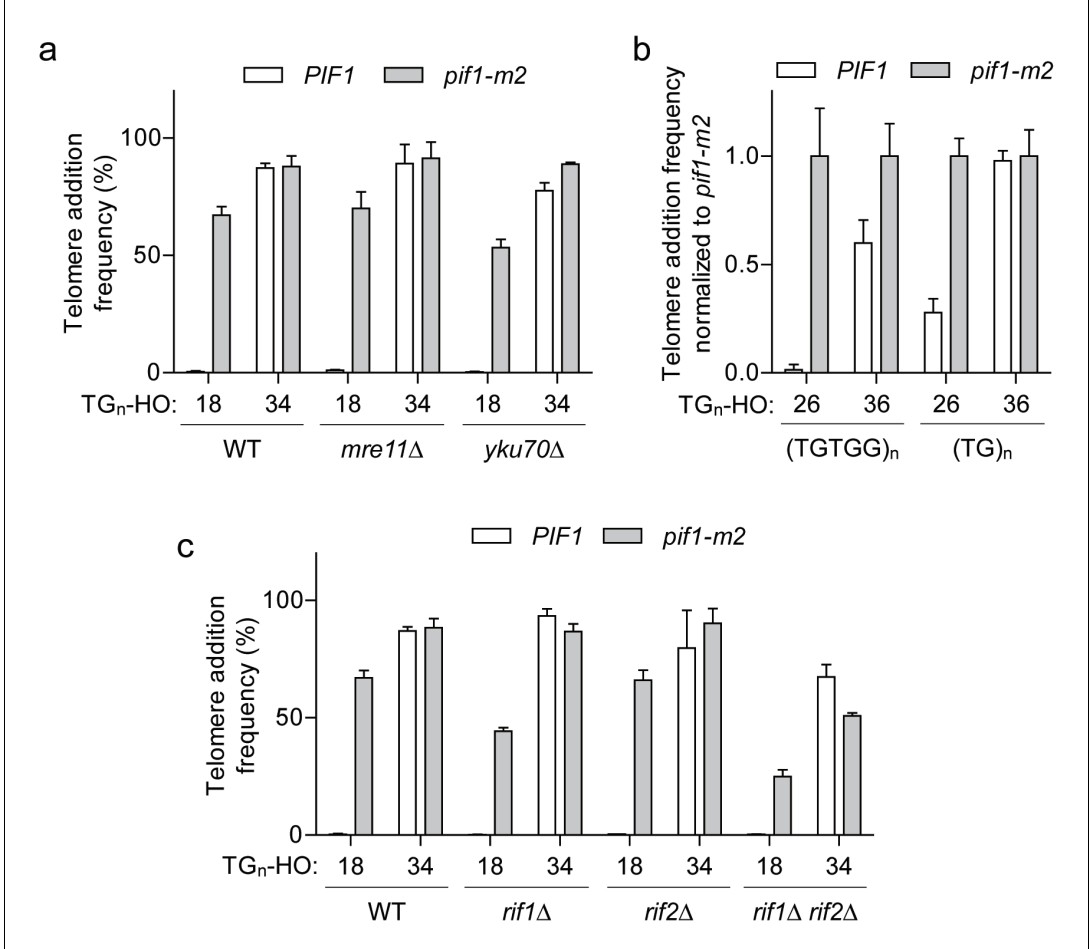

**Figure 5.** The DSB-telomere transition does not require Rap1. (a) Telomere addition frequency at the $TG_{18}$ and $TG_{34}$ DNA ends in *mre11Δ* and *yku70Δ* mutants. Data represent the mean ± s.d. from n = 3 independent experiments. Source data are found in *Figure 5—source data 1*. (b) Telomere addition frequency normalized to *pif1-m2* cells at DNA ends containing 26 bp or 36 bp of either $(TGTGG)_n$ or $(TG)_n$ repeats. Data represent the mean ± s.d. from n = 3 independent experiments. Source data are found in *Figure 5—source data 2*. (c) Telomere addition frequency at the $TG_{18}$ and $TG_{34}$ DNA ends in *rif1Δ*, *rif2Δ*, and *rif1Δ rif2Δ* double mutants. Data represent the mean ± s.d. from n = 3 independent experiments. The WT data in panels (a) and (c) are the same as that shown in *Figure 3c*. Source data are found in *Figure 5—source data 3*.
DOI: https://doi.org/10.7554/eLife.23783.027

The following source data is available for figure 5:

**Source data 1.** Raw data for telomere addition assays shown in *Figure 5a*.
DOI: https://doi.org/10.7554/eLife.23783.028

**Source data 2.** Raw data for telomere addition assays shown in *Figure 5b*.
DOI: https://doi.org/10.7554/eLife.23783.029

**Source data 3.** Raw data for telomere addition assays shown in *Figure 5c*.
DOI: https://doi.org/10.7554/eLife.23783.030

The loss of either complex, however, did not affect either side of the DSB-telomere transition (*Figure 5a*; source data are available in *Figure 5—source data 1*).

Budding yeast telomeres are bound by two specialized proteins, Rap1 and Cdc13, and the binary nature of the DSB-telomere transition suggests that the discrete binding of either protein may trigger insensitivity to Pif1. As Cdc13 binds ssDNA at the distal end of the telomere (*Lin and Zakian, 1996*; *Nugent et al., 1996*), an attractive prediction is that Rap1 bound to double-stranded telomeric DNA of longer TG repeats might inhibit Pif1. This model nicely correlates with the observed length of the DSB-telomere transition, as Cdc13 and Rap1 bind DNA sequences of 11 bp (*Hughes et al., 2000*) and 18 bp, respectively (*Gilson et al., 1993*; *Ray and Runge, 1999*). Rap1 has also been previously shown to stimulate telomere addition (*Grossi et al., 2001*; *Lustig et al., 1990*; *Ray and Runge, 1998*).

Rap1 is an essential protein that binds the consensus DNA sequence of 5′-ACA<u>CCC</u>ATACACC-3′ containing an invariable CCC core (*Wahlin and Cohn, 2000*; *Grossi et al., 2001*; *Graham and Chambers, 1994*). Importantly, substitution of the middle cytosine to guanine in this motif abolishes Rap1 binding (*Grossi et al., 2001*; *Graham and Chambers, 1994*). To test whether Rap1 is required to bypass Pif1 activity at DNA ends, we first generated synthetic telomeric sequences with strict $(TGTGG)_n$ or $(TG)_n$ repeats in both 26 bp and 36 bp lengths. Unlike natural telomeres, both sequences lack a CCC motif on the opposing strand. Despite these alterations, we still observed increased telomere addition at $TG_{36}$ ends in wild-type cells (*Figure 5b*; source data are available in *Figure 5—source data 2*), suggesting that Rap1 binding is not required for this phenomenon.

As telomere length regulation by Rap1 is coordinated through two downstream negative regulators of telomerase, Rif1 and Rif2 (*Levy and Blackburn, 2004*; *Wotton and Shore, 1997*), we asked whether these proteins are important for the DSB-telomere transition. Consistent with a Rap1-independent mechanism, telomere addition at the $TG_{34}$ end was unaltered in *rif1Δ rif2Δ* mutants (*Figure 5c*; source data are available in *Figure 5—source data 3*).

## Cdc13 function influences the fate of DNA ends

The Cdc13 N-terminal OB-fold domain (OB1) (*Figure 6a*) forms dimers (*Mitchell et al., 2010*; *Sun et al., 2011*) and in vitro can also bind telomeric ssDNA repeats of 37 and 43 bp, but not 18 and 27 bp (*Mitchell et al., 2010*), neatly matching our observed threshold. We hypothesized that Cdc13 dimerization and its unique N-terminal binding mode might allow longer DNA ends to bypass Pif1 and sought to test this idea by disrupting dimerization with the *cdc13-L91A* mutation (*Mitchell et al., 2010*). Consistent with this prediction, telomere addition at the $TG_{34}$ end was inhibited by Pif1 in *cdc13-L91A* cells (*Figure 6b*; source data are available in *Figure 6—source data 1*); however, further investigation revealed a growth defect in these mutants that was suppressed by *pif1-m2* (*Figure 6c*). This result is reminiscent of the defective *cdc13-1* allele, which is also suppressed by loss of *PIF1* (*Downey et al., 2006*; *Addinall et al., 2008*). High copy plasmid expression of *cdc13-L91A* was able to rescue the growth defect, but also increased telomere addition at the $TG_{34}$ substrate (*Figure 6b*) arguing that the initially observed defect in *cdc13-L91A* mutants was not solely due to impaired N-terminal dimerization.

We next performed a mutagenesis screen to identify *CDC13* alleles that have become sensitive to Pif1 activity (*Figure 6d,e*). Screening of approximately 6000 mutants led to the identification of fifteen hits that exhibited impaired telomere addition at the $TG_{34}$ substrate. As this screen was performed in wild-type cells, we next determined if the mutations could support telomere addition in the absence of Pif1. Recovered plasmids were re-transformed into wild-type and *pif1-m2* cells, and analysis of telomere addition revealed two clones with minor phenotypes (#7 and 14), five clones with reduced telomere addition in both wild-type and *pif1-m2* cells (#37, 48, 79, 80, and 81), and eight clones in which telomere addition was impaired in wild-type cells but relatively unaffected in *pif1-m2* cells (#1, 2, 3, 42, 63, 71, 72, and 77) (*Figure 6f*; source data are available in *Figure 6—source data 2*). This observation suggested that the third group of Cdc13 mutations specifically sensitize the $TG_{34}$ end to the activity of Pif1 and are herein referred to as *cdc13-sp* alleles (<u>s</u>ensitive to <u>P</u>if1).

DNA sequencing revealed an average of 11 amino acid substitutions per *cdc13-sp* allele and methodical mapping experiments led to the identification of causative amino acid substitutions in six of the eight *cdc13-sp* mutants (*Table 1*, highlighted in red). Three alleles had contributions from multiple substitutions: I87N and Y758N in *cdc13-sp1*, H12R and F728I in *cdc13-sp72*, and E566V,

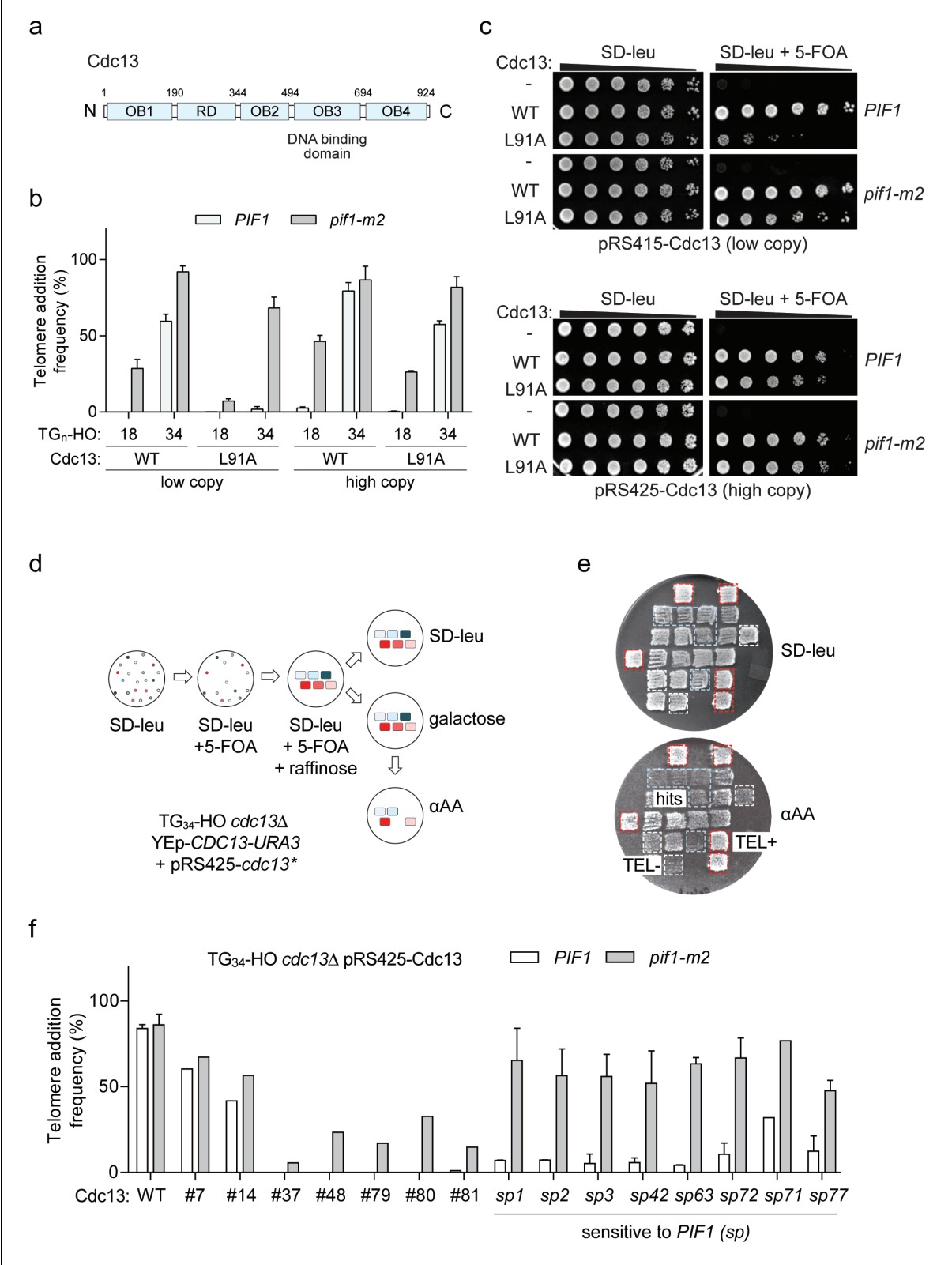

**Figure 6.** A genetic screen to identify Cdc13 mutants that prevent telomere addition at the $TG_{34}$ end. (**a**) Schematic of Cdc13 domain architecture consisting of four OB-fold domains (OB1-4) and a telomerase recruitment domain (RD). (**b**) Telomere addition frequency at the $TG_{18}$ and $TG_{34}$ DNA ends in *cdc13Δ* cells expressing wild-type *CDC13* (WT) or *cdc13-L91A* from a low copy (pRS415) or high copy (pRS425) plasmid. Data represent the mean ± s.d. from n = 3 independent experiments. Source data are found in *Figure 6—source data 1*. (**c**) Spot assays to determine cell viability in
*Figure 6 continued on next page*

*Figure 6 continued*

cdc13Δ cells with a covering YEp-CDC13-URA3 plasmid and pRS415- or pRS425-derived plasmids expressing wild-type Cdc13 (WT) or Cdc13-L91A. Fivefold serial dilutions of yeast cultures were grown on SD-leu as a control, and on SD-leu +5 FOA to determine viability in the absence of the covering plasmid. Plates were grown at 30°C for 2–3 days. (d) Schematic of a screen in TG$_{34}$ cdc13Δ cells using a plate-based genetic assay for telomere addition. Repaired mutant cdc13 plasmids were selected on SD-leu and the covering YEp-CDC13-URA3 removed by plating on 5-FOA before DSB induction. This step also eliminates all inviable cdc13 mutations. Plates were incubated for 2–3 days at 30°C with the exception of galactose plates which were incubated for 4 hr. An agar plate was used to reduce cell number before final selection. (e) Example of re-testing plate from the screen. Cdc13 mutants that prevent telomere addition are identified by the inability to grow on media containing α-aminoadipate (α-AA) (blue box), compared to positive control wild-type cells which add telomeres (red box) and TG$_{18}$ cells that do not (white box). (f) Telomere addition frequency at the TG$_{34}$ DNA end in PIF1 and pif1-m2 cells in a cdc13Δ background expressing recovered pRS425-Cdc13 mutants from the screen. Data represent the mean ± s.d. from n = 1 experiment for hits #7–81, and n = 2 independent experiments for all cdc13-sp alleles. Source data are found in *Figure 6— source data 2*.

DOI: https://doi.org/10.7554/eLife.23783.031

The following source data is available for figure 6:

**Source data 1.** Raw data for telomere addition assays shown in *Figure 6b*.
DOI: https://doi.org/10.7554/eLife.23783.032

**Source data 2.** Raw data for telomere addition assays shown in *Figure 6f*.
DOI: https://doi.org/10.7554/eLife.23783.033

N567D, and Q583K in cdc13-sp3 (*Figure 7a*; source data are available in *Figure 7—source data 1*). Cdc13-I87, like L91, is also implicated in OB1 dimerization (*Mitchell et al., 2010*), again hinting that disrupting this function may restore Pif1 activity. The moderate telomere addition defect of Cdc13-I87N was likely only identified in the screen due to further exacerbation by the Y758N mutation (*Figure 7a*). The most important mutation in cdc13-sp3 was identified to be Q583K with a minor contribution from E556V/N567D. Interestingly, all three residues are found in the canonical DNA-binding domain, suggesting that weakening the association of Cdc13 with telomeric DNA can also sensitize the TG$_{34}$ end to Pif1.

Three single amino acid residues (F236, S255, Q256) could completely account for the phenotype of the remaining three alleles (cdc13-sp63, cdc13-sp42, and cdc13-sp77, respectively; *Figure 7a*). These residues all map to the Cdc13 telomerase recruitment domain, suggesting that weakening the association of Cdc13 with telomerase is another means to facilitate Pif1 activity at TG$_{34}$. In particular, the S255A mutation has previously been shown to impair telomerase recruitment, resulting in telomere shortening (*Gao et al., 2010*; *Tseng et al., 2006*). Similarly, the classic telomerase null cdc13-2 (E252K) mutant (*Nugent et al., 1996*) was also sensitive to Pif1 (*Figure 7b*; source data are available in *Figure 7—source data 2*). Telomere length in several other cdc13-sp alleles was also reduced in both wild-type and pif1-m2 backgrounds, and the severity of the defect generally correlated with the magnitude of the telomere addition phenotype (*Figure 7c*).

The diversity of Cdc13 mutations that sensitize the TG$_{34}$ end to Pif1 (*Figure 7d*) suggests that generally disrupting Cdc13 function facilitates Pif1 activity by shifting the balance away from telomere addition. In agreement with this idea, the cdc13-1 mutant grown at permissive temperature was also sensitive to Pif1, (*Figure 7b*); its P371S mutation is now known to disrupt OB2 dimerization (*Mason et al., 2013*). Furthermore, analysis of hits from our screen that decreased telomere addition in both wild-type and pif1-m2 cells revealed double mutations of critical residues including S255L/ Q256R in clone 37, I87T/F236Y in clone 40, and F236S/E252K in clone 48, suggesting that strongly disrupting Cdc13 function eventually impairs telomere addition even in the absence of PIF1. In line with this idea, the F235S/E252K/Q583K triple mutant prevented telomere addition at a TG$_{34}$ end even in pif1-m2 cells (*Figure 7b*).

The striking effect of the cdc13 recruitment domain mutants led us to explore in greater detail how disrupting the ability of Cdc13 to recruit telomerase influences the DSB-telomere transition. We determined the telomere addition frequency of the cdc13-Q256H mutant on a series of DNA ends of varying telomere length (*Figure 7e*; source data are available in *Figure 7—source data 3*). While telomere addition frequency was very low at TG$_{34}$, it gradually increased as the telomere sequence lengthened. Thus, mutating the recruitment domain of CDC13 abolished the sharp DSB-telomere transition and increased the length of telomere sequence needed to allow telomere addition to become resistant to Pif1 action.

**Table 1.** Mutations in *cdc13-sp* alleles.

pRS425-*cdc13*\* plasmids were recovered from cells grown on SD-leu media that were unable to grow on α-AA containing media. *Cdc13* mutations were identified by plasmid sequencing. Mutations highlighted in red were identified by mapping experiments to determine which amino acid substitutions contribute to the mutant phenotype. Mutations highlighted in blue target important Cdc13 residues identified in this study or in previous work (*Nugent et al., 1996*; *Lendvay et al., 1996*), which are predicted to contribute to the defect, although these exact substitutions were not specifically tested.

| Allele | Mutations | | | | | | | |
|---|---|---|---|---|---|---|---|---|
| cdc13-sp1 | Y27F | I87N | S175P | D322G | L386M | T733A | S737C | Y758N |
| cdc13-sp3 | Q220K | L242P | E566V | N567D | Q583K | K695R | | |
| cdc13-sp42 | Q36R | F58L | L131S | D150G | K161I | S170A | N194D | V217I |
| | S228T | S255P | K329E | L362I | F389L | | | |
| cdc13-sp63 | F236S | V396I | F539Y | E716G | T756P | I767V | A807T | P896S |
| cdc13-sp72 | H12R | F96L | K129N | L179S | T291A | K296E | N426S | K469R |
| | E566G | E570G | I648N | F728I | P896S | | | |
| cdc13-sp77 | T3P | V38A | D102G | K135N | N240Y | Q256H | E264D | T266S |
| | S288C | I346V | D430G | S467R | N470S | S490A | M498V | K618R |
| | E636V | H687R | L721M | V725L | D792N | T779A | | |
| cdc13-sp2 | R83K | P101L | I174F | N180S | E197G | G243E | V367L | G404A |
| | S494P | R503G | L571R | I594M | E679G | | | |
| cdc13-sp71 | H168R | I247N | E252G | T291P | V424I | K504R | F587L | T710S |
| | Y816H | | | | | | | |
| clone 37 | Q66R | F96L | E121K | F142L | S255L | Q256R | I342T | N378D |
| | E416A | L425F | L452M | | | | | |
| clone 40 | I87T | F236Y | Y626F | F665Y | T907S | | | |
| clone 48 | F58S | T112S | F187I | F236S | E252K | A280V | D601A | S643P |
| clone 79 | I72V | K73R | Q94L | E192G | D219G | V238A | Q256H | K296E |
| | G325R | K365I | K469R | R495G | I523T | H777Q | | |
| clone 80 | N14K | Y70H | I72F | L436F | F575L | | | |
| clone 81 | E121V | N180D | N199D | A231S | F236Y | M258N | M276T | G295R |
| | S314N | I366F | I412V | N455I | M525V | M579V | M625I | E716G |
| | D773V | K909E | | | | | | |

DOI: https://doi.org/10.7554/eLife.23783.038

## Discussion

The work presented here sheds light on how cells distinguish between DSBs and short telomeres and reveals a sharp transition in the fate of DNA ends with regards to their sensitivity to the telomerase inhibitor Pif1. Our findings agree with previous reports demonstrating that linear plasmid substrates containing 41 bp of telomeric repeats are efficiently converted into telomeres (*Lustig, 1992*). We find that the DSB-telomere transition also exists at natural chromosome ends and that Cdc13 is a key player in setting this transition.

The observed behavior of Pif1 complements several known mechanisms that tightly integrate telomeric sequence length and the regulation of telomerase. The identified activity of Pif1 at telomeric repeats under 34 bp joins a Mec1-dependent mechanism that inhibits Cdc13 binding at repeats under 11 bp (*Zhang and Durocher, 2010*), highlighting the importance of inhibiting telomerase at DSBs. Conversely, we propose that DNA ends containing telomeric sequences of 34 bp to 125 bp are recognized as critically short telomeres and are preferentially elongated. Tel1 is implicated as a key regulator in this process (*Chang et al., 2007*; *Sabourin et al., 2007*; *Hector et al., 2007*; *Arnerić and Lingner, 2007*; *Cooley et al., 2014*), although the exact phosphorylation targets are unknown (*Gao et al., 2010*). Finally, the canonical counting mechanism of telomeres is known to

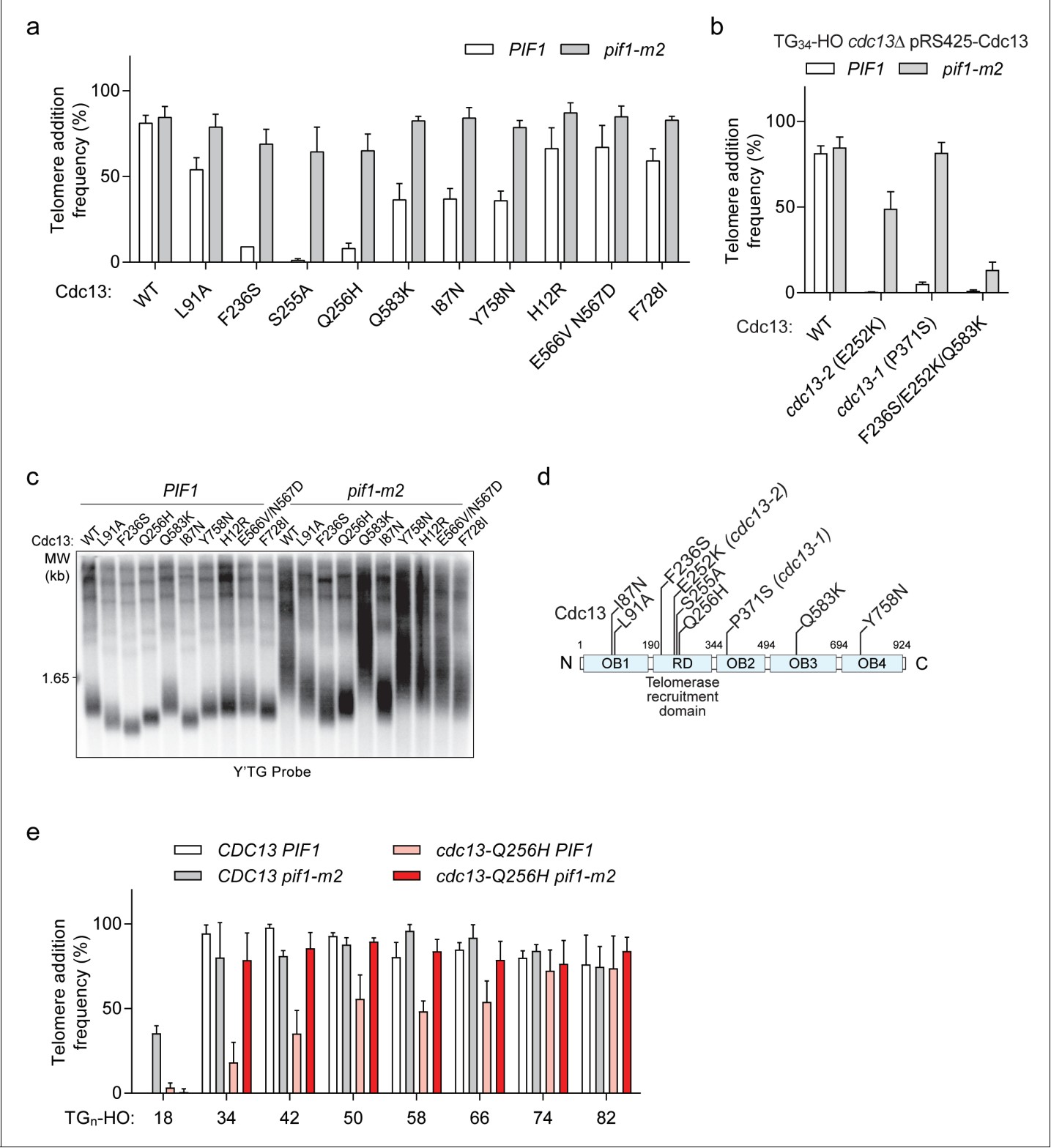

**Figure 7.** Cdc13 mutations that sensitize the $TG_{34}$ end to Pif1 activity. (**a**) Telomere addition frequency at the $TG_{34}$ DNA end in *PIF1* and *pif1-m2* cells in a *cdc13Δ* background expressing wild-type or mutated Cdc13 from pRS425. Data represent the mean ± s.d. from n = 3 independent experiments. Source data are found in *Figure 7—source data 1*. (**b**) Telomere addition frequency at the $TG_{34}$ DNA end in *PIF1* and *pif1-m2* cells in a *cdc13Δ* background expressing plasmid-borne wild-type or mutated Cdc13. The *cdc13-1* mutant was grown at a permissive temperature of 23°C. Data represent the mean ± s.d. from n = 3 independent experiments. Source data are found in *Figure 7—source data 2*. (**c**) Southern blot for telomere

*Figure 7 continued on next page*

*Figure 7 continued*

length of the strains examined in panel (a). Cells were passaged for approximately 75 generations before genomic DNA extraction. A Y' TG probe was used to label telomere sequences. (d) Schematic of Cdc13 domain architecture, with the most significant *cdc13-sp* amino acid mutations indicated. The majority of these mutations lie within the telomerase recruitment domain. (e) Telomere addition frequency at DNA ends containing 18–82 bp of TG sequence. Data represent the mean ± s.d. from a minimum of n = 3 independent experiments. Source data are found in *Figure 7—source data 3*.
DOI: https://doi.org/10.7554/eLife.23783.034
The following source data is available for figure 7:

**Source data 1.** Raw data for telomere addition assays shown in *Figure 7a*.
DOI: https://doi.org/10.7554/eLife.23783.035
**Source data 2.** Raw data for telomere addition assays shown in *Figure 7b*.
DOI: https://doi.org/10.7554/eLife.23783.036
**Source data 3.** Raw data for telomere addition assays shown in *Figure 7e*.
DOI: https://doi.org/10.7554/eLife.23783.037

limit the extension of long telomeres through the negative regulators Rif1 and Rif2 (*Marcand et al., 1997*; *Levy and Blackburn, 2004*; *McGee et al., 2010*; *Hirano et al., 2009*). Remarkably, although short telomeres of 34 bp or longer are insensitive to Pif1, Pif1 preferentially binds and acts at wild-type length or longer telomeres, thereby helping to promote the elongation of short telomeres (*Phillips et al., 2015*).

In order to maintain genome stability, the length of telomeric repeat sequence necessary to over-come Pif1 should be greater than any natural sequence occurring within the genome. Any longer sequences should therefore be prone to conversion into new telomeres and might be under nega-tive selection during evolution due to the loss of genetic material. Consistent with this idea, the two longest $(TG_{1-3})_n$ sequences in the correct orientation outside of telomeric regions in budding yeast include a 35 bp sequence on Chr VII, and a 31 bp sequence on Chr VI (*Mangahas et al., 2001*).

Our investigation into the molecular trigger of the DSB-telomere transition points to a key role for the DNA-binding protein Cdc13. This conclusion is supported by work revealing that microsatel-lite repeats containing Cdc13-binding sites stimulate telomere addition (*Piazza et al., 2012*), and recent data that a hotspot on Chr V also promotes Cdc13 binding and telomere addition (*Obodo et al., 2016*). Furthermore, the tethering of Cdc13, but not Rap1, to this site was shown to be sufficient for the formation of new telomeres (*Obodo et al., 2016*). Moreover, resection at DNA ends with short stretches of telomere repeats may remove all potential Rap1-binding sites, strongly suggesting that the DSB-telomere transition depends either on ssDNA-binding proteins like Cdc13 or the ssDNA itself.

The ability of the Cdc13 OB1 domain to dimerize and bind DNA provides an attractive solution to the DSB-telomere transition; however, our results clearly indicate that sensitivity to Pif1 is not unique to any one domain and can result from a variety of mutations throughout Cdc13, most nota-bly in the recruitment domain. Weakening the ability of Cdc13 to recruit telomerase provides a satis-fying explanation for the sensitivity of the $TG_{34}$ end to Pif1 (*Figure 8*) but is unable to explain why the $TG_{34}$ end is resistant to Pif1 in the first place, especially as fusing telomerase to Cdc13 was unable to overwhelm Pif1 at the $TG_{18}$ substrate. Interestingly, the mammalian CST complex can bind single-stranded telomeric DNA of 32 bp and longer (*Miyake et al., 2009*) suggesting that Cdc13 in combination with Stn1 and Ten1 may also possess unique binding properties.

One key unresolved issue is the mechanism by which Pif1 inhibits telomerase on either side of the DSB-telomere transition, and our results with the *pif1-4A* and *-4D* alleles suggest that these activities may be distinct. It is clear that Pif1 can remove telomerase RNA from telomeres (*Boulé et al., 2005*; *Li et al., 2014*), but genetic data reveal that Pif1 also has telomerase-independent activity as *PIF1* loss increases the growth of *cdc13-1 tlc1Δ* cells (*Dewar and Lydall, 2010*). One potential activity for Pif1 at DSBs is through the promotion of DNA end resection, first observed in *cdc13-1* mutants (*Dewar and Lydall, 2010*). Consistent with this possibility, end resection impairs telomere addition, and new telomeres are added closer to DSB sites in *pif1-m2* cells (*Chung et al., 2010*). This model therefore predicts that the $TG_{18}$ end may be resected with the help of Pif1, but that resection is blocked at the $TG_{34}$ end, thus providing an explanation as to why tethering telomerase to the $TG_{18}$ end does not increase telomere addition. In line with this prediction, a $TG_{22}$ end was previously

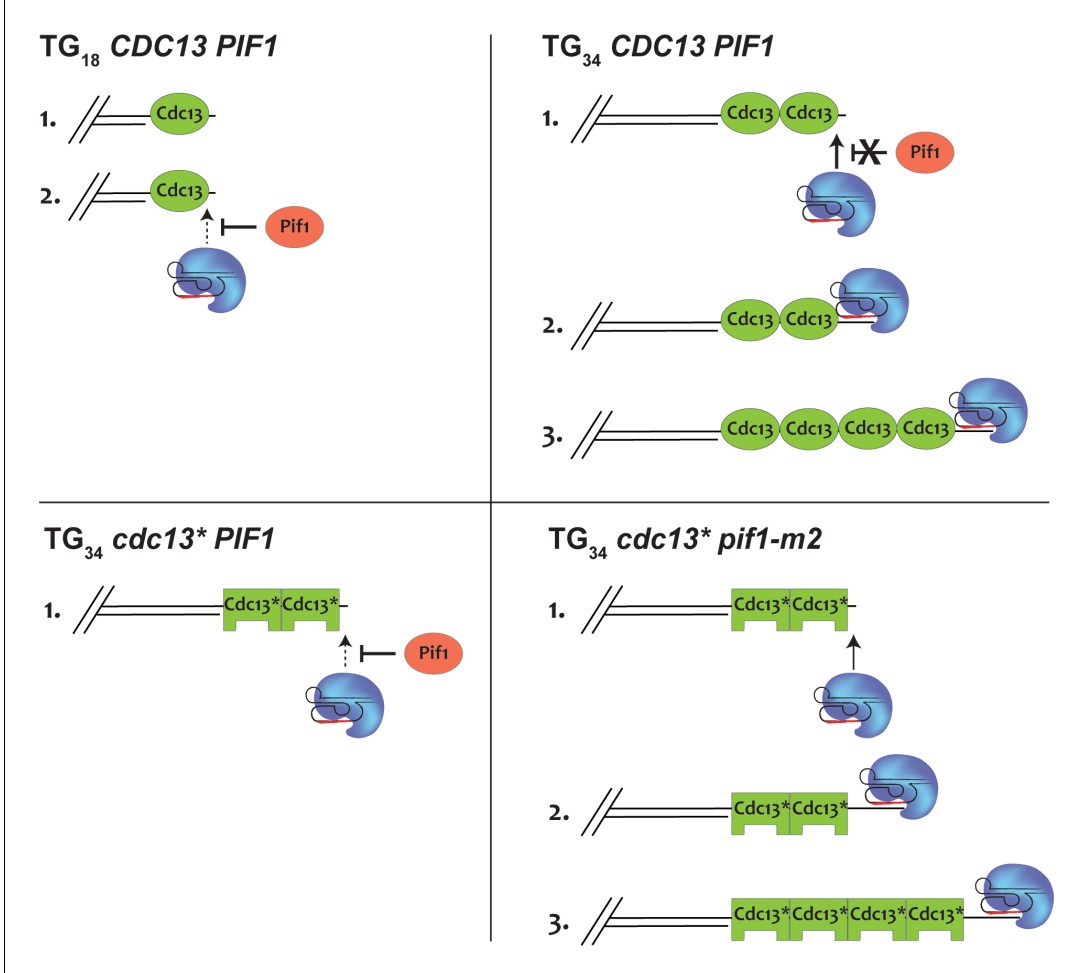

**Figure 8.** A model to explain how Cdc13 and Pif1 cooperate to establish the DSB-telomere transition. At the $TG_{18}$ end, Pif1 is able to efficiently prevent telomerase-mediated extension (top left). At the $TG_{34}$ end, Cdc13-mediated recruitment of telomerase overpowers Pif1 activity, allowing the extension of the end (top right). This is impaired if the recruitment domain of Cdc13 is mutated (bottom left). Further mutation of *PIF1* restores the ability of telomerase to elongate the $TG_{34}$ end (bottom right).

DOI: https://doi.org/10.7554/eLife.23783.039

observed to partially suppress DNA end resection compared to a $TG_{11}$ substrate (*Hirano and Sugimoto, 2007*).

In conclusion, using Pif1 as a cellular indicator for the DNA-end fate decision reveals a striking threshold that recapitulates several properties of DSBs and telomeres. We propose that a $TG_{34}$ DNA end, which is approximately a tenth of the size of a healthy budding yeast telomere, is interpreted by the cell as a minimal telomere.

## Materials and methods

### Yeast strain construction and growth

The genotypes of the yeast strains used in this study are listed in *Supplementary file 1b*. Strains were constructed by standard allele replacement, PCR-mediated gene deletion or epitope-tagging methods, or via transformations of the indicated plasmids. The desired mutations were selected by prototrophy or drug selection and verified by PCR or sequencing. Standard yeast media and growth conditions were used (*Treco and Lundblad, 2001*; *Sherman, 2002*).

Telomeric repeats were cloned into the pVII-L plasmid which features an HO endonuclease cut site, a *URA3* selection marker, and homology arms for integration at the *ADH4* locus

(*Gottschling et al., 1990*). Longer telomeric repeats were assembled using commercial gene synthesis (Mr. Gene, Regensberg, Germany) while Quikchange mutagenesis (Agilent, Santa Clara, CA) was performed for further manipulation of repeat sequences. Insertions and deletions of up to 30 bp of TG repeats were robustly obtained in a single round of mutagenesis. Quikchange-mediated shortening of a large $TG_{250}$ sequence also yielded a wide range of shorter repeats. All repeats were verified by DNA sequencing before integration.

The $TG_{82}$-HO cassette on Chr VII was replaced by integrating SalI and EcoRI-digested pVII-L plasmids and selecting for colonies on SD-ura. Single integration of the plasmid and HO cleavage at the locus was confirmed by Southern blot. Telomere addition strains were constructed in a *rad52Δ* background with a covering pRS414-Rad52 plasmid to facilitate genome manipulation through homologous recombination. Strains were cured of the pRS414-Rad52 plasmid by random loss in non-selective media and colonies were screened by replica-plating to SD-trp.

To make SSY76, the *GALL* promoter was amplified from plasmid pYM-N27 (*Janke et al., 2004*) using primers EST1_S1 (5′-GAAAAAGTATATTCCATTAAATGACACATGCCA CCATAGATAATGCG TACGCTGCAGGTCGAC-3′) and EST1_S4 (5′-CTTGAAAAATAATCTCATACATTCTTCGTTAACTTC TTCATTATCCATCGATGAATTCTCTGTCG-3′). Correct insertion was verified by PCR with EST1.1 (5′-CAGACGAAGGTGCTTTCA-3′) and EST1.4 (5′-GCTCTTCGAGAAACCTAG-3′) primers.

Pif1 mutations were generated by Quikchange mutagenesis on a pAUR101-*pif1-m1* nuclear-specific construct and integrated at the *AUR1* locus in *pif1-m2* cells. The *est2-up34* mutation was generated by pop-in/pop-out gene replacement.

## Telomere addition assays

Telomere addition assays were performed as previously described (*Zhang and Durocher, 2010*). Briefly, yeast cultures were grown overnight in XY (10 g/L yeast extract, 20 g/L bactopeptone, 0.1 g/L adenine, 0.2 g/L tryptophan) + 2% glucose to log phase and subcultured into XY + raffinose (2%) for overnight growth to a density of $2.5–7.5 \times 10^6$ cells/mL. Nocodazole (Sigma Aldrich, Oakville, Canada) was added at 15 µg/mL for 2 hr to synchronize cells in G2/M before addition of galactose to induce HO endonuclease expression. Cells were plated on XY +glucose plates before the addition of galactose and 4 hr after galactose addition, and grown for 2–3 days. The total number of colonies were counted, following which colonies were replica-plated to media containing α-aminoadipic acid (α-AA; Sigma-Aldrich) to identify cells that had lost the distal *LYS2* gene on Chr VII. Frequency of telomere addition was calculated as the percent of post-galactose surviving colonies that were α-AA resistant. An alternative calculation, (α-AA resistant colonies/ (pre-galactose colonies - α-AA sensitive colonies)), revealed the same threshold of Pif1 activity between the $TG_{18}$ and $TG_{34}$ ends, but with increased variability between experiments.

## Genomic DNA extraction

Genomic DNA was isolated using a phenol-chloroform extraction protocol. Briefly, overnight cultures of cells were grown to saturation, pelleted, and resuspended with 200 µL 'Smash and Grab' lysis buffer (10 mM Tris-Cl, pH 8.0, 1 mM EDTA, 100 mM NaCl, 1% SDS, 2% Triton X-100). 200 µL of glass beads (Sigma Aldrich, 400–600 µm diameter) were added along with 200 µL phenol-chloroform (1:1). Cells were lysed by vortexing for 5 min before addition of 200 µL TE buffer (10 mM Tris-Cl pH 8, 1 mM EDTA). Samples were centrifuged at 4°C and DNA from the upper layer precipitated with the addition of 1 mL ice-cold 100% ethanol and centrifuged at 4°C. The DNA pellets were resuspended in 200 µL TE with 300 µg RNAse A (Sigma-Aldrich) and incubated at 37°C for 30 min. DNA was again precipitated with the addition of 1 mL ice-cold 100% ethanol and 10 µL of 4 M ammonium acetate, centrifuged, dried, and resuspended in TE.

## Southern blots for telomere addition and length

Fifteen micrograms of genomic DNA were digested overnight with SpeI (for $TG_{82}$ strains) or EcoRV (for all other TG repeat lengths). Digested DNA was run on a 1% agarose gel in 0.5X TBE buffer (45 mM Tris-borate, 1 mM EDTA) at 100 V for 6 hr, denatured in the gel for 30 min with 0.5 M NaOH and 1.5 mM NaCl, and neutralized for 30 min with 1.5 M NaCl and 0.5 M Tris-Cl pH 7.5. DNA was transferred to Hybond N + membrane (GE Healthcare Life Sciences, Mississauga, Canada) using overnight capillary flow and 10X SSC buffer (1.5 M NaCl, 150 mM sodium citrate, pH 7). Membranes

were UV-crosslinked (Stratalinker 1800, Agilent) and blocked at 65°C with Church hybridization buffer (250 mM NaPO$_4$ pH 7.2, 1 mM EDTA, 7% SDS). Radiolabeled probes complementary to the *ADE2* (for TG$_{82}$ strains) or *URA3 gene* (for all other TG repeat lengths) were generated from purified PCR products using the Prime-It Random labeling kit (Agilent) and $\alpha^{32}$-dCTP. Membranes were probed overnight, washed three times with 65°C Church hybridization buffer and exposed overnight with a phosphor screen (GE Healthcare Life Science) before imaging on a Storm or Typhoon FLA 9000 imager (GE Healthcare Life Sciences). Quantification of the added telomere signal (above CUT band) was performed in ImageQuant (GE Healthcare Life Sciences) by subtracting the background signal before HO induction followed by normalization to the internal loading control (INT). Telomere length analysis was performed by digesting genomic DNA with XhoI and probing with a Y'-TG probe generated from the pYT14 plasmid (*Shampay et al., 1984*) or with a telomere-specific (5'-CACCA-CACCCACACACCACACCCACA-3') probe.

## Inducible STEX (iSTEX) assay

The *tlc1-tm* allele was amplified from MCY415 using primers oSMS1 (5'-ACCTGCCTTTGCAGATCC TT-3') and TLC1-RV (5'-TTATCTTTGGTTCCTTGCCG-3'). The obtained product was transformed into SSY76 cells, which were then plated onto YPD +G418 plates. Genomic DNA of several independent transformants was prepared using a Wizard Genomic DNA Purification Kit (Promega, Leiden, Netherlands). The *TLC1* locus was again amplified using primers oSMS1 and TLC1-RV and sequenced using primer oSMS2 (5'-TGTAGATGCTTGTGTGTG-3') to confirm proper integration of the mutant *tlc1-tm* allele. Overnight cultures of *tlc1-tm* transformants were inoculated, diluted to OD$_{600}$ = 0.1 in 25 mL YPD the next morning, and diluted again at the end of the day to OD$_{600}$ = 0.0005 in 100 mL YPD, so that the cultures would be in log phase the morning after. Cells were then arrested with 0.045 μg/mL alpha factor (Sigma-Aldrich) for 1 hr in YPD at 30°C, spun down, resuspended in YP media containing 2% galactose and 0.0225 μg/mL alpha factor and incubated for 2 hr at 30°C. Next, cells were spun down and washed several times, resuspended in YPGal with 50 μg/mL of pronase E (Sigma-Aldrich), and cultured for 2 hr at 30°C. Cells were harvested at various points during the experiment for flow cytometry analysis and telomere PCR.

## Flow cytometry

Cells were fixed in 70% ethanol overnight at 4°C, washed with demineralized H$_2$O (dH$_2$O) and incubated in 50 mM Tris-Cl pH 8 containing 0.1 mg/mL RNase A (Thermo Fisher Scientific, Landsmeer, Netherlands, cat. no. EN0531) for 2–4 hr at 37°C. They were then spun down and resuspended in 50 mM Tris-Cl pH 7.5 containing 0.1 mg/mL proteinase K (Sigma-Aldrich, cat. no. 3115801001) and incubated for 30–60 min at 50°C. Cells were next resuspended in FACS buffer (200 mM Tris-Cl pH 7.5, 200 mM NaCl, 78 mM MgCl$_2$), incubated with SYTOX Green Nucleic Acid Stain (Thermo Scientific, cat. no. S7020) in 50 mM Tris pH 7.5, and sonicated at high intensity (3 cycles of 30 s on and 30 s off) before analysis.

## Telomere PCR

Telomere V-R and VI-R PCR was performed essentially as described (*Förstemann et al., 2000*; *Chang et al., 2007*). 1 μL of genomic DNA (~100 ng) was mixed with 8 μL of 1x NEBuffer 4 (New England Biolabs (NEB), Ipswich, MA) and boiled for 10 min at 94°C. 1 μL of tailing mix (0.05 μL Terminal Transferase (NEB, cat. no. M0315), 0.1 μL 10x NEBuffer 4, 0.1 μL 10 mM dCTP, 0.75 μL dH$_2$O) was added and incubated for 30 min at 37°C, 10 min at 65°C, and 5 min at 96°C. Immediately after tailing, 30 μl of PCR mix was added. The PCR mix consisted of 4 μL 10x PCR buffer (670 mM Tris-HCl pH 8.8, 160 mM (NH$_4$)$_2$SO$_4$, 50% glycerol, 0.1% Tween-20), 0.32 μL 25 mM dNTP mix, 0.3 μL 100 μM telomere-specific primer (V-R: 5'-GTGAGCGGATAACAATTTCACACAGTCTAGATG TCCGAATTGATCCCAGAGTAG-3' or VI-R: 5'-ACGTGTGCGTACGCCATATCAATATGC-3'), 0.3 μL 100 μM G$_{18}$ primer (5'-CGGGATCCG$_{18}$-3'), 0.5 μL Q5 High-Fidelity DNA Polymerase (NEB, cat. no. M0491), 24.68 μL dH$_2$O. The samples were denatured at 98°C for 3 min, followed by 35 cycles of 98°C for 30 s and 68°C for 15 s, and a final extension step at 72°C for 2 min.

## Gel extraction, cloning, and sequencing

Telomere PCR products were separated on 2.5% agarose gels and extracted using a NucleoSpin®
Gel and PCR Clean-up kit (Macherey-Nagel, Düren, Germany, cat. no. 740609). The purified PCR
products were then cloned using a Zero Blunt TOPO PCR Cloning Kit (Thermo Fisher Scientific, cat.
no. 450245). Individual clones were sequenced by GATC Biotech (Cologne, Germany) and the result-
ing data were analyzed using Sequencher software (Gene Codes, Ann Arbor, MI).

## PCR mutagenesis screens

Mutant alleles were generated by error-prone PCR using Taq polymerase (New England Biolabs)
and 0.25 mM MnCl, and purified using spin columns (Qiagen, Mississauga, Canada). The Pif1 muta-
genesis screen was performed in $TG_{82}pif1-m2$ cells co-transformed with gapped pRS416-$pif1-m1$
and purified inserts. Cells harbouring repaired plasmids were selected on SD-ura. The Cdc13 muta-
genesis screen was performed in $TG_{34}cdc13\Delta$ cells containing a covering YEp-$CDC13$-$URA3$ plasmid
and co-transformed with gapped pRS425-$CDC13$ plasmid and PCR inserts. Cells harboring repaired
plasmids were selected on SD-ura before replica-plating to 5-fluoroorotic acid (5-FOA) to remove
the covering plasmid. Mutant $cdc13$ alleles that are defective in capping should be inviable at this
step. Colonies from both screens were patched onto raffinose plates and grown for 2 days before
replica plating to galactose plates for 4 hr, and finally to $\alpha$-AA plates after reducing cell density by
first replicating plating to a blank agar plate. Plasmids were rescued using a phenol-chloroform
extraction and transformed into *Escherichia coli*. Plasmids were sequenced to identify mutations and
retransformed into the parental yeast strain to confirm that the phenotype resulted from the plasmid
mutation.

## Statistics

The statistics carried out in *Figure 2* were done using a Fisher's exact test and for this analysis, telo-
meres containing only wild-type divergence were excluded.

## Acknowledgements

We are grateful to Katrin Paeschke and members of the Durocher laboratory for critical reading of
the manuscript. JS is supported by a CIHR Doctoral award. DD is the Thomas Kierans Chair in Mech-
anisms of Cancer Development and a Canada Research Chair (Tier 1) in the Molecular Mechanisms
of Genome Integrity. Work in the MC laboratory was supported by a Vidi grant from the Nether-
lands Organization for Scientific Research, and work in the DD laboratory was supported by CIHR
grant (FDN143343) and a Grant-in-Aid from the Krembil Foundation.

## Additional information

### Funding

| Funder | Grant reference number | Author |
| --- | --- | --- |
| Canadian Institutes of Health Research | | Jonathan Strecker<br>Daniel Durocher |
| Krembil Foundation | | Daniel Durocher |
| Nederlandse Organisatie voor Wetenschappelijk Onderzoek | | Sonia Stinus<br>Michael Chang |
| Canadian Institutes of Health Research | FDN143343 | Daniel Durocher |

The funders had no role in study design, data collection and interpretation, or the
decision to submit the work for publication.

### Author contributions

Jonathan Strecker, Conceptualization, Investigation, Writing—original draft, Writing—review and
editing; Sonia Stinus, Investigation, Writing—review and editing; Mariana Pliego Caballero, Formal

analysis, Investigation, Writing—review and editing; Rachel K Szilard, Resources, Data curation, Validation, Investigation, Writing—review and editing; Michael Chang, Resources, Supervision, Funding acquisition, Methodology, Writing—original draft, Project administration, Writing—review and editing; Daniel Durocher, Conceptualization, Resources, Supervision, Funding acquisition, Methodology, Writing—original draft, Project administration, Writing—review and editing

### Author ORCIDs
Michael Chang  http://orcid.org/0000-0002-1706-3337
Daniel Durocher  http://orcid.org/0000-0003-3863-8635

### Decision letter and Author response
Decision letter https://doi.org/10.7554/eLife.23783.042
Author response https://doi.org/10.7554/eLife.23783.043

## Additional files

### Supplementary files
• Supplementary file 1. Description of telomere sequences and yeast strains used in this study.
DOI: https://doi.org/10.7554/eLife.23783.040

• Transparent reporting form
DOI: https://doi.org/10.7554/eLife.23783.041

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
