## [Decision Letter]

Thank you for submitting your article "A sharp Pif1-dependent threshold separates DNA double-strand breaks from critically short telomeres" for consideration by *eLife*. Your article has been reviewed by three peer reviewers, and the evaluation has been overseen by Reviewing Editor Kathleen Collins and Kevin Struhl as the Senior Editor. The following individuals involved in review of your submission have agreed to reveal their identity: Teresa Teixeira (Reviewer #3); Dorothy Shippen (Reviewer #4).

The Reviewing Editor has drafted this decision to help you prepare a revised submission.

Summary:

Cells must distinguish short telomeres from DNA double strand breaks to preserve genome stability, yet the mechanism that underlies the distinction remains poorly understood. Recent studies implicate the Pif1 helicase as a pivotal player in determining the fate of these two types of chromosome termini. Here the authors investigate the mechanism by which Pif1 from *S. cerevisiae* blocks de novo telomere formation on DSBs with less than 26bp of telomeric DNA, but not at tracts of longer telomere sequence. Using a classical HO-induced DSB system, and a single molecule telomere length assay, the authors establish a telomere length "window" of ~26-40bp that distinguishes DSBs from short telomeres.

To examine how Pif1 activity is repressed at telomeric DNA sequences that exceed this length threshold, they use a genetic approach to test the contributions of DDR kinases (Mec1, Tel1 and Rad53), telomerase recruitment, and Pif1 phosphorylation. They find no correlation with Pif1 activity at telomeres longer than the "DSB-telomere transition length". Similarly, Rap1 binding does not significantly impact Pif1 activity. However, the authors find a correlation between various aspects of Cdc13 function and Pif1 activity at telomeres longer than 34bp. Using a suppressor screen they identify numerous residues in Cdc13 important for suppressing Pif1 activity.

This study is important for its discovery of a link between Cdc13 and Pif1 in distinguishing DSBs from short telomeres. Overall the study was solid technically and communicated clearly, with a few omissions detailed below. The major shortcoming of the work is that it doesn't satisfyingly resolve the nature of the link between Cdc13 and Pif1 in the context of previously known properties/roles of Cdc13. The impact of the study could be considerably strengthened by additional mechanistic insight about how Cdc13 modulates Pif1 activity at different DNA ends.

Essential revisions:

1) Major experiments using the HO cut assay are elegant and well controlled. However, the evidence that what is learned applies to native telomeres is less strong because it lacks the positive control in *pif1-m2* background. This control would be ideal to show that absence of telomerase action at critically short telomeres is not due to some transient cell cycle arrest due checkpoint activation or cell death in the (small) portion of cells containing the exact same critically short telomeres we are looking at. Note that despite differences in the experimental settings, STEX is done 2h after telomerase re-expression, while in Southerns shown for the HO cut we see no telomere elongation at this time point after Gal addition to induce HO. This is challenging (need to construct an inducible system for Pif1 to prevent telomeres from lengthening before the experiment), but it would help in more confidently extend the conclusions to native ends. Alternatively, STEX could be tried in mutants of MEC1 or RAD53, but it's not obvious that these mutations wouldn't affect telomerase action at these very short telomeres (see effect at TG82-HO in Figure 3—figure supplement 1). Anyway, authors should try to provide evidence that the portion of cells carrying the critically short telomeres contain active telomerase.

2) The STEX result in Figure 2 (reduced telomere addition to very short endogenous telomeres) is at odds with a number of published STEX results that don't show this effect. How do the authors explain this discrepancy? How do the authors eliminate the possibility that use of a mutant template sequence alters the results? I do not understand the derivation of the p value given for the VI-R telomeres. The Fisher Exact Test gives a p value of 0.07 (75 extended and 159 not extended among the longer telomeres vs. 0 extended and 7 not among the shortest telomeres). How do the authors justify using a different cut-off length in the two cases?

3) The authors state in the Materials and methods that the total cell counts both pre- and post-galactose treatment are obtained, but the fraction of cells that survive the 4-hour treatment with galactose is not given. Could the authors comment whether this rate differs greatly between situations in which the end either is or is not efficiently healed by telomere addition?

4) In several experiments, the authors fail to see an effect on TG34-HO healing by expression of a *pif1-m1* allele in a *pif1-m2* strain (Figure 3 for the 5AQ allele and Figure 4 for the 4D allele). The authors should verify that the mutant Pif1 protein is expressed and functional by virtue of its ability to complement the telomere lengthening phenotype of the *pif1-m2* allele.

5) If "WT" results are presented in multiple figures, the legend must clearly state that data are identical to those shown previously. See Figure 3, Figure 4, Figure 5, and 5C. Are the same *sml1* and *sml1 pif1-m2* data shown in panels B and D of Figure 3—figure supplement 1? What is the n value for these samples?

6) In Figure 6, the authors should show the effect of high copy expression in the TG18 PIF1 and *pif1-m2* strains. Low copy Cdc13 does not fully complement since telomere addition at TG18 in the *pif1-m2* background is less than half of what is seen in other experiments. Does overexpression of WT Cdc13 render TG18 partially or fully resistant to Pif1 action?

7) In Figure 8, the effect of the cdc13-S255A mutation is particularly strong, but this mutant is not included on the telomere length analysis of Figure 8. Why? If a mutation like S255A raises the threshold (in terms of telomere length) required for an end to become resistant to Pif1 action, one might predict that longer telomeric seeds will regain that ability. It would be informative to repeat the experiment in Figure 1 with one of the stronger cdc13 alleles (or at least to check one of the longer TG strains) to test this idea.

8) In the Discussion, the authors provide only a cursory mention of nucleolytic resectioning at the HO cut site prior to telomere addition. The DNA substrate that arises following HO cleavage will be grossly distinct from a natural short telomere, but this is glossed over in the text. Because resectioning is not discussed, the rationale for looking at the contribution of the double strand telomere binding protein RAP1 versus the single strand telomere protein Cdc13 is not established. Clearly, the 3' overhang depicted in Figure 1 is not consistent with the major conclusions from the paper. A more accurate description of the actual substrate for action at the "DSB-telomere transition" is required.

9) In the last paragraph of the subsection “Identification of a Pif1-insensitivity threshold at DNA ends”, a major conclusion of this paper is there is a sharp boundary for the DSB-telomere transition length between 26 to 34bp. Figure 1 provides evidence to support this conclusion. However, with the TG30 sequence there is a 30% addition rate, arguing that the addition frequency actually gradually increases inside this window. Additional experiments that test Pif1 activity on TG28 and TG32 are needed to determine whether there is indeed a sharp length threshold or a more diffuse window. Furthermore, according to the hypothesis that Pif1 regulation is mediated by Cdc13 binding to DNA and that three Cdc13 binding sites are needed for regulation (11bp each), then both TG30 and TG26 should be insensitive to Pif1 activity since neither contain three complete Cdc13 binding sites.

10) The suppressor screen is powerful, but the data provide little molecular insight into how Cdc13 impedes Pif1 function. Mutations localize in the vicinity of different functional domains of Cdc13. To emphasize this point, the authors should provide a diagram (not a table) of residues identified in the suppressor screen to show their position within Cdc13 relative to known functional elements (dimerization, telomerase recruitment, DNA binding). The authors begin to dissect the contribution of such functional domains (e.g. dimerization) by assessing the cdc13-L91A mutation in their telomere addition assay. In addition, they should directly evaluate that contribution Cdc13 DNA binding and telomerase recruitment on Pif1 regulation using other biochemically validated mutants. This is essential because the authors have not provided any evidence that the residues they identified in their genetic screen are in fact important for specific Cdc13 functions. Consequently, we are left with no clear hypothesis for how Cdc13 regulates Pif1.

---

## [Author Response]

*Essential revisions:*

*1) Major experiments using the HO cut assay are elegant and well controlled. However, the evidence that what is learned applies to native telomeres is less strong because it lacks the positive control in pif1-m2 background. This control would be ideal to show that absence of telomerase action at critically short telomeres is not due to some transient cell cycle arrest due checkpoint activation or cell death in the (small) portion of cells containing the exact same critically short telomeres we are looking at. Note that despite differences in the experimental settings, STEX is done 2h after telomerase re-expression, while in Southerns shown for the HO cut we see no telomere elongation at this time point after Gal addition to induce HO. This is challenging (need to construct an inducible system for Pif1 to prevent telomeres from lengthening before the experiment), but it would help in more confidently extend the conclusions to native ends. Alternatively, STEX could be tried in mutants of MEC1 or RAD53, but it's not obvious that these mutations wouldn't affect telomerase action at these very short telomeres (see effect at TG82-HO in Figure 3—figure supplement 1). Anyway, authors should try to provide evidence that the portion of cells carrying the critically short telomeres contain active telomerase.*

The reviewers raise an important point. We have completed the *pif1-m2* iSTEX controls (see revised Figure 2 and Figure 2—figure supplement 1). At telomere VR, there is a statistically significant increase in the extension of telomeres below the DSB-telomere boundary in *pif1-m2* cells. A similar result is seen for telomere VI-R, but the number of short telomeres was not sufficient to reach statistical significance (p=0.151) despite our best efforts to accrue the number of short telomeres. Nevertheless, our new findings strengthen our conclusion that there is a telomereDSB transition at native telomeres that is regulated by Pif1.

*2) The STEX result in Figure 2 (reduced telomere addition to very short endogenous telomeres) is at odds with a number of published STEX results that don't show this effect. How do the authors explain this discrepancy? How do the authors eliminate the possibility that use of a mutant template sequence alters the results? I do not understand the derivation of the p value given for the VI-R telomeres. The Fisher Exact Test gives a p value of 0.07 (75 extended and 159 not extended among the longer telomeres vs. 0 extended and 7 not among the shortest telomeres). How do the authors justify using a different cut-off length in the two cases?*

This comment can be subdivided in a number of points. First, regarding the comparison of our results with those of the literature, we note that there are only four papers that show STEX data (Teixeira et al., 2004; Arnerić and Lingner, 2007; Ji et al., 2004; Phillips et al., 2015). The Philips et al. paper is not highly relevant to this comment because there was a high background of telomerase-independent divergence events at telomeres below 50 nt in length. The other papers did indeed show some extension events at telomeres below our 30-40 nt threshold, but their sample size was quite small in each of these studies. However, it is critical to note that since those studies used strains with wild-type telomerase, it is simply impossible to differentiate between telomerase-dependent and -independent events, and therefore it is impossible to assess how many of the divergence events observed in the previous papers were due to telomerase activity.

As for the second point regarding the impact of the telomerase template, we contend that it is unlikely that the mutant template alters the iSTEX results for the following reasons. First, the incipient telomeres are wild-type and we only express the mutant telomerase for a single cycle. Second, previous work (Chang et al., 2007) alongside the present work here show no difference in terms of repeat addition processivity, frequency of extension, length of extension, and preferential elongation of short telomeres by the mutant template-containing telomerase. This important point is now emphasized in the first paragraph of the subsection “A DSB-telomere transition also exists at chromosome ends”.

With respect to the third point, the reviewers were correct and we had indeed miscalculated the p value in the original Figure 2. During the revision period, the number of sequenced telomeres accrued and a new p value is now reported (p=0.039).

As for the last point, the difference in the DSB-telomere threshold for telomere V-R (43 nt) and telomere VI-R (37 nt) is likely due to experimental variation and/or the sample size of telomere ends analyzed by iSTEX. We contend that the 6-nt difference is a rather small difference. In addition, the exact sequences of these telomeres are different, and this may impact somewhat the threshold length.

*3) The authors state in the Materials and methods that the total cell counts both pre- and post-galactose treatment are obtained, but the fraction of cells that survive the 4-hour treatment with galactose is not given. Could the authors comment whether this rate differs greatly between situations in which the end either is or is not efficiently healed by telomere addition?*

The fraction of cells that survive the 4-h treatment with galactose varies from 10% to 40%, and this variation cannot be easily explained by genotype or TG length. We believe this variation could be due to differences in HO-cutting efficiency from experiment to experiment. This difference does not alter in any way the interpretation of our results. Whether we calculate telomere addition frequency from the total number of cells or from only the surviving cells, TG18 is sensitive to PIF1 status while TG34 is not (Author response image 1).

**Author response image 1. respfig1:** Telomere addition frequency at DNA ends containing 18 or 34 bp of TG sequence normalized either to the total amount of cells (total) or only the surviving cells after the 4h galactose induction (surviving). Data represent the mean ± s.d. from a minimum of n=3 independent experiments.

*4) In several experiments, the authors fail to see an effect on TG34-HO healing by expression of a pif1-m1 allele in a pif1-m2 strain (Figure 3 for the 5AQ allele and Figure 4 for the 4D allele). The authors should verify that the mutant Pif1 protein is expressed and functional by virtue of its ability to complement the telomere lengthening phenotype of the pif1-m2 allele.*

This is a good point. We have now included a new experiment (revised Figure 3—figure supplement 1) that shows, as expected, that the *pif1-m1* allele rescues the telomere lengthening phenotype of the *pif1-m2* allele.

*5) If "WT" results are presented in multiple figures, the legend must clearly state that data are identical to those shown previously. See Figure 3, Figure 4, Figure 5, and 5C. Are the same sml1 and sml1 pif1-m2 data shown in panels B and D of Figure 3—figure supplement 1? What is the n value for these samples?*

We apologize for the oversight. The mentioned figures (3C, 4E, 5A, 5C) did indeed show the same WT data. The telomere addition of *sml1∆* cells is also the same in panels B and D of Figure 3—figure supplement 1 (n=3). The figure legends have been modified to explicitly mention this.

*6) In Figure 6, the authors should show the effect of high copy expression in the TG18 PIF1 and pif1-m2 strains. Low copy Cdc13 does not fully complement since telomere addition at TG18 in the pif1-m2 background is less than half of what is seen in other experiments. Does overexpression of WT Cdc13 render TG18 partially or fully resistant to Pif1 action?*

This is a good suggestion. As requested, we have tested the impact of high-copy expression of *CDC13* and *cdc13-L91A* in the *PIF1* and *pif1-m2* strains (revised Figure 6). Overexpression of wild-type *CDC13* clearly does not render TG_18_ resistant to Pif1 action and therefore the differences seen between low- and highcopy expression was specific for *cdc13-L91A*.

*7) In Figure 8, the effect of the cdc13-S255A mutation is particularly strong, but this mutant is not included on the telomere length analysis of Figure 8. Why? If a mutation like S255A raises the threshold (in terms of telomere length) required for an end to become resistant to Pif1 action, one might predict that longer telomeric seeds will regain that ability. It would be informative to repeat the experiment in Figure 1 with one of the stronger cdc13 alleles (or at least to check one of the longer TG strains) to test this idea.*

The *cdc13-S255A* mutation has previously been shown to have short telomeres by two independent groups (Tseng et al., 2006; Gao et al., 2010). This has now been noted in the fourth paragraph of the subsection “Cdc13 function influences the fate of DNA ends” and the work are cited. To answer the second part of the reviewers’ comments, we have tested the *cdc13-Q256H* mutant across a range of TG substrates (see new Figure 8). As predicted, we find that the *cdc13-Q256H* mutant has the ability to elongate longer TG substrates. Interestingly, the threshold is no longer as sharp as in the wild-type situation, but instead is more gradual.

*8) In the Discussion, the authors provide only a cursory mention of nucleolytic resectioning at the HO cut site prior to telomere addition. The DNA substrate that arises following HO cleavage will be grossly distinct from a natural short telomere, but this is glossed over in the text. Because resectioning is not discussed, the rationale for looking at the contribution of the double strand telomere binding protein RAP1 versus the single strand telomere protein Cdc13 is not established. Clearly, the 3' overhang depicted in Figure 1 is not consistent with the major conclusions from the paper. A more accurate description of the actual substrate for action at the "DSB-telomere transition" is required.*

The reviewers raise an interesting point and we agree that the resection of DNA ends would preclude Rap1 from binding in the first place. We should have mentioned the impact of resection on short telomeric substrates in the original Discussion. Furthermore, since the G-tails of very short telomeres also likely remove Rap1 binding sites, ssDNA formation at DSBs and critically short telomeres further suggests that Cdc13 rather than Rap1 regulates the DSB-telomere transition. We now mention this possibility in the revised Discussion.

The reviewers should also note that the DNA substrate depicted in Figure 1 is the structure produced immediately after HO cleavage. This was noted in the figure legend and it was never intended to suggest that this was the substrate at the time of telomerase action. Since extensive resection inhibits de novo telomere addition, one could predict that the exact structure of the cleaved TG_34+_-HO substrates would be similar, to some extent, to that of a short telomere with a G-tail.

*9) In the last paragraph of the subsection “Identification of a Pif1-insensitivity threshold at DNA ends”, a major conclusion of this paper is there is a sharp boundary for the DSB-telomere transition length between 26 to 34bp. Figure 1 provides evidence to support this conclusion. However, with the TG30 sequence there is a 30% addition rate, arguing that the addition frequency actually gradually increases inside this window. Additional experiments that test Pif1 activity on TG28 and TG32 are needed to determine whether there is indeed a sharp length threshold or a more diffuse window. Furthermore, according to the hypothesis that Pif1 regulation is mediated by Cdc13 binding to DNA and that three Cdc13 binding sites are needed for regulation (11bp each), then both TG30 and TG26 should be insensitive to Pif1 activity since neither contain three complete Cdc13 binding sites.*

Although the word “sharp” is obviously subjective, when viewed in the context of a full-length telomere (~300 bp), we believe the transition window (~8 bp) from a Pif1sensitive to a Pif1-insensitive end is quite narrow. To further highlight this point, when the DSB-transition is altered by mutation of *CDC13*, the transition is much more gradual (see revised Figure 8). In addition, the TG_30_ telomere addition rate (as shown in Figure 1) is 19% (normalized to *pif1-m2*), not 30%.

While it is true that the minimum length for Cdc13 binding was determined to be 11 bp, there is no evidence to suggest that three Cdc13 molecules would bind a 33 bp telomere sequence. Indeed, steric hindrance could well prevent this from occurring. To avoid confusion, these sentences have now been removed from the text.

*10) The suppressor screen is powerful, but the data provide little molecular insight into how Cdc13 impedes Pif1 function. Mutations localize in the vicinity of different functional domains of Cdc13. To emphasize this point, the authors should provide a diagram (not a table) of residues identified in the suppressor screen to show their position within Cdc13 relative to known functional elements (dimerization, telomerase recruitment, DNA binding). The authors begin to dissect the contribution of such functional domains (e.g. dimerization) by assessing the cdc13-L91A mutation in their telomere addition assay. In addition, they should directly evaluate that contribution Cdc13 DNA binding and telomerase recruitment on Pif1 regulation using other biochemically validated mutants. This is essential because the authors have not provided any evidence that the residues they identified in their genetic screen are in fact important for specific Cdc13 functions. Consequently, we are left with no clear hypothesis for how Cdc13 regulates Pif1.*

As requested, we now included in the revised manuscript a diagram of Cdc13 with the strongest mutations indicated (revised Figure 8). The S255A mutant has previously been carefully studied and shown to be defective in telomerase recruitment (Gao et al., 2010). We have now included telomere addition assays for the *cdc13-2* (E252K) mutant, which is also known to be defective in telomerase recruitment (revised Figure 8).